# Quantitative trait and transcriptome analysis of genetic complexity underpinning cardiac interatrial septation in mice using an advanced intercross line

**Mahdi Moradi Marjaneh[1,2,3]\*†, Edwin P Kirk[1,4,5]†, Ralph Patrick[1]†, Dimuthu Alankarage[1], David T Humphreys[1,2], Gonzalo Del Monte-Nieto[1,2], Paola Cornejo-Paramo[1], Vaibhao Janbandhu[1,2], Tram B Doan[1], Sally L Dunwoodie[1,2], Emily S Wong[1,6], Chris Moran[7], Ian CA Martin[7], Peter C Thomson[7], Richard P Harvey[1,2,6]\***

[1]Victor Chang Cardiac Research Institute, Sydney, Australia; [2]School of Clinical Medicine, Faculty of Medicine and Health, UNSW Sydney, Sydney, Australia; [3]Department of Infectious Disease, Faculty of Medicine, Imperial College London, London, United Kingdom; [4]Centre for Clinical Genetics, Sydney Children's Hospital, Sydney, Australia; [5]Randwick Genomics Laboratory, NSW Health Pathology, Sydney, Australia; [6]School of Biotechnology and Biomolecular Sciences, Faculty of Science, UNSW Sydney, Sydney, Australia; [7]Sydney School of Veterinary Science, University of Sydney, Sydney, Australia

**\*For correspondence:**
m.moradi@imperial.ac.uk (MMM);
r.harvey@victorchang.edu.au (RPH)

†These authors contributed equally to this work

**Competing interest:** The authors declare that no competing interests exist.

**Abstract** Unlike single-gene mutations leading to Mendelian conditions, common human diseases are likely to be emergent phenomena arising from multilayer, multiscale, and highly interconnected interactions. Atrial and ventricular septal defects are the most common forms of cardiac congenital anomalies in humans. Atrial septal defects (ASD) show an open communication between the left and right atria postnatally, potentially resulting in serious hemodynamic consequences if untreated. A milder form of atrial septal defect, patent *foramen ovale* (PFO), exists in about one-quarter of the human population, strongly associated with ischaemic stroke and migraine. The anatomic liabilities and genetic and molecular basis of atrial septal defects remain unclear. Here, we advance our previous analysis of atrial septal variation through quantitative trait locus (QTL) mapping of an advanced intercross line (AIL) established between the inbred QSi5 and 129T2/SvEms mouse strains, that show extremes of septal phenotypes. Analysis resolved 37 unique septal QTL with high overlap between QTL for distinct septal traits and PFO as a binary trait. Whole genome sequencing of parental strains and filtering identified predicted functional variants, including in known human congenital heart disease genes. Transcriptome analysis of developing septa revealed downregulation of networks involving ribosome, nucleosome, mitochondrial, and extracellular matrix biosynthesis in the 129T2/SvEms strain, potentially reflecting an essential role for growth and cellular maturation in septal development. Analysis of variant architecture across different gene features, including enhancers and promoters, provided evidence for the involvement of non-coding as well as protein-coding variants. Our study provides the first high-resolution picture of genetic complexity and network liability underlying common congenital heart disease, with relevance to human ASD and PFO.

## Editor's evaluation

Overall, this is a comprehensive study that will provide a useful reference for the field. It will be a useful tool for hypothesis generation, which could lead to research on therapies that target atrial septal or common congenital heart disease.

## Introduction

Congenital heart disease (CHD) is common with a neonatal incidence of 0.8–1%, placing an enormous burden on affected patients, families, and healthcare systems (*van der Linde et al., 2011*). High throughput sequencing has identified abundant examples of monogenic syndromic and non-syndromic forms of CHD; however, the majority represent complex multifactorial conditions of unknown etiology, with a significant role for polygenic disease, de novo mutations, and gene-environment interactions (*Sifrim et al., 2016*). Genome-wide association studies (GWAS) have identified a small number of common CHD risk alleles, likely those of the largest effect (*Lahm et al., 2021*); however, for the majority of clinical subcategories, none have been identified thus far.

Development of the four-chambered mammalian heart leads to the permanent separation of the systemic and pulmonary circulations through septation of common atrial and ventricular chambers. Septation co-evolved with air-breathing and involves the convergence of myogenic and non-myogenic tissues at the valvuloseptal apparatus of the atrioventricular (AV) junction (*Moorman and Christoffels, 2003*). This process is genetically vulnerable as septation defects occur commonly in the human CHD spectrum (*Gruber and Epstein, 2004*).

During fetal life, the inter-atrial septum functions initially as a one-way 'flap valve' to help divert systemic blood away from the pulmonary circulation (*Figure 1A*). Regulatory networks generated at the boundaries between different cardiac progenitor fields define the programs for septal development (*De Bono et al., 2018*; *Rana et al., 2014*; *Steimle et al., 2018*). Initially, a muscular *septum primum* grows inwards from the atrial roof (*Anderson et al., 2003*) associated with a mesenchymal cap at its leading edge. The *septum primum* cap and an additional mesenchyme at its base called the dorsal mesenchymal protrusion (*Burns et al., 2016*; *Webb et al., 1998*), then fuse with mesenchyme at the AV complex (*Deepe et al., 2020*). Before its closure, the upper edge of the *septum primum* becomes fenestrated by apoptosis, forming a left-right communication termed the *ostium secundum* (*Anderson et al., 2003*; *Moore and Persaud, 1998*). A *septum secundum* forms as an infolding of the dorsal atrial wall in humans (*Anderson et al., 2003*) or as a specific muscular ridge in mice (*Briggs et al., 2012*), leaving an additional, prominent and offset interatrial communication termed the *foramen ovale* (*Burns et al., 2016*), completing the flap valve apparatus (*Figure 1A*).

When the lungs become activated at birth, left atrial pressure increases and the flap valve normally closes permanently by fusion of the *septum primum* to the *septum secundum* (*Figure 1B*, left panel). However, fusion is incomplete in about one-quarter of the human population, leading to the condition termed *patent foramen ovale* (PFO; *Figure 1B*, right panel) (*Hagen et al., 1984*). For larger PFO, there is a strong probability of a hemodynamically significant inter-atrial communication associated with a higher risk of cryptogenic (unexplained) stroke, likely due to the passage of venous thrombi across the patent septum to the systemic circulation (*Lechat et al., 1988*; *Webster et al., 1988*). Larger PFOs are also associated with atrial septal aneurysm (ASA) (*Hagen et al., 1984*; *Homma et al., 2003*), as well as migraine with aura, clinical hypoxemia, and decompression illness in divers (*Shnaider et al., 2004*; *Torti et al., 2004*; *Wilmshurst et al., 2000*).

Atrial septal defect (ASD) is a less prevalent but more severe abnormality of the atrial septum (*Feldt et al., 1971*; *Figure 1B*, middle panel). There is likely a genetic link and anatomical continuum between the more common *secundum* form of ASD (ASDII) with PFO and ASA (*Kirk et al., 2006*; *Kirk et al., 2007*; *Posch et al., 2010*). ASDII arises from abnormalities of the *septum primum* and/or *secundum*, and presents as a frank and permanent interatrial corridor (*Zipes, 2005*), and, if untreated, the left-to-right blood shunt present postnatally can cause pulmonary hypertension and Eisenmenger syndrome, a life-threatening complication (*Beghetti and Galiè, 2009*).

Analysis of QTL has emerged as an approach to understand the genetic complexity underpinning both quantitative and complex (non-Mendelian) binary traits (*Ma et al., 2020*; *Shirai and Okada, 2021*). Our previous study of inbred mouse strains revealed significant variation in atrial septal anatomy correlating with PFO and ASA (*Biben et al., 2000*), with a genetic background as a major determinant.

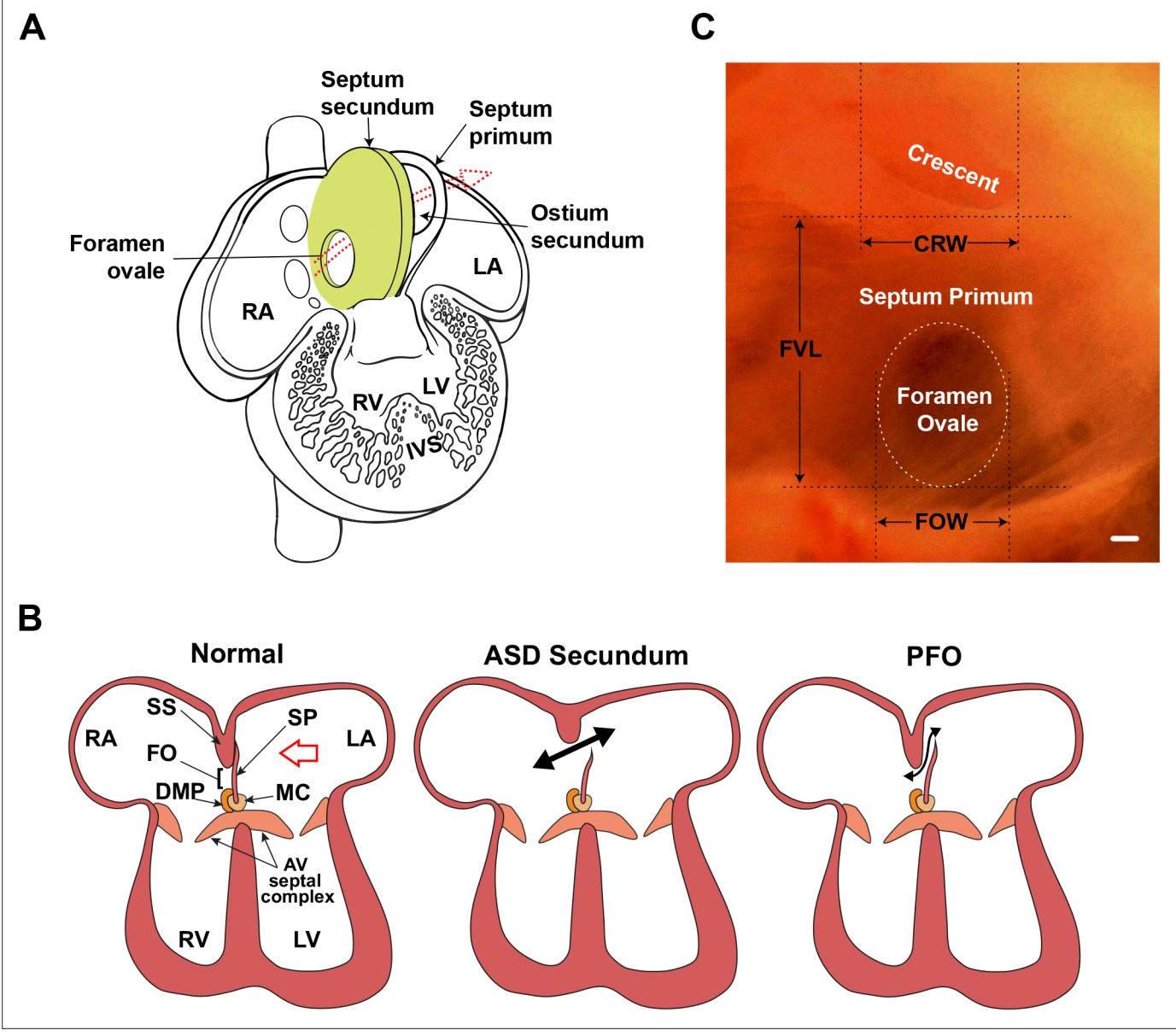

**Figure 1.** Atrial septal morphology in normal adult and congenital heart disease. (**A**) Schematic of human fetal heart highlighting inter-atrial septal structures. In the fetal heart, the *septum primum* has not yet fused with the *septum secundum,* as in adult stages; thus, the *foramen ovale* is patent. The *foramen ovale* and *ostium secundum* are offset, creating a flap valve. Arrow shows the direction of the blood shunt, allowing blood to bypass the pulmonary circuit. (**B**) Schematic of atrioventricular septal complex in the adult heart, and dysmorphology associated with the more common secundum type of atrial septal defect (ASD), and patent *foramen ovale* (PFO). Red arrow in the normal heart indicates the viewpoint for panel **C**. Double-headed black arrows indicate blood flow in pathological settings. (**C**) Light micrograph of the interatrial septum of an adult mouse heart as seen after removal of the left atrial appendage. The annulus of the mitral valve is towards the lower left of this panel. Note that the *foramen ovale* is covered by the membranous atrial *septum primum*, which is foremost in the left viewpoint (see red arrow in panel **B**). Septal landmarks are shown along with quantitative atrial septal traits measured in F2 and F14 studies. The crescent corresponds to the upper edge (see **A**) of the atrial *septum primum*. Scale bar, 100 μm. AV: atrioventricular; CRW: crescent width; DMP: dorsal mesenchymal protrusion; FO: *foramen ovale*; FOW: *foramen ovale* width; FVL: flap valve length; IVS: interventricular septum; LA: left atrium; LV: left ventricle; MC: mesenchymal cap; RA: right atrium; RV: right ventricle; SP: *septum primum*; SS: *septum secundum*.

© 2001, Elsevier. Figure 1A has been adapted from Figure 7-16 from *Larsen, 1997*, with permission from Elsevier. It is not covered by the CC-BY 4.0 license and further reproduction of this panel would need permission from the copyright holder.

The online version of this article includes the following figure supplement(s) for figure 1:

**Figure supplement 1.** Histograms of quantitative traits in F14 mice with and without patent *foramen ovale* (PFO).

We established quantitative parameters of septal status including length of the *septum primum* (flap valve length; FVL), orthogonal width of the *foramen ovale* (FOW), and width of the open corridor in PFO (crescent width; CRW) (*Figure 1B and C*). We found that mean FVL was strongly negatively correlated with the prevalence of PFO across a variety of genetic backgrounds (*Biben et al., 2000*) and, collectively, short FVL, large FOW, and large CRW were all strongly associated with PFO risk (*Kirk et al., 2006*). All traits were influenced by mutation of the cardiac homeodomain transcriptional master-regulator NKX2-5, pathogenic variants in which are strongly causative for ASDII in humans (*Biben et al., 2000*; *Schott et al., 1998*). We performed a QTL analysis with an F2 design using QSi5 and 129T2/SvEms strains, representing mice with extremes of atrial septal phenotypes, identifying seven significant (logarithm of odds (LOD) >4.3) and six suggestive (LOD >2.8) QTL affecting quantitative septal traits (*Kirk et al., 2006*), indicating a complex genetic basis for atrial septation defects in inbred mice. The F2 design used in our previous study has the power to detect the most significant QTL underlying complex traits, but confidence intervals are usually large, and each may reflect the effects of multiple loci and can contain hundreds of potential candidate genes (*Darvasi et al., 1993*). Therefore, increasing recombination has become the focus of fine mapping approaches including the use of advanced intercross lines (AIL), generated by intercrossing inbred strains with extreme phenotypes across 10 or more generations to increase chromosomal recombination (*Darvasi and Soller, 1995*). A similar rationale underlies the establishment of multi-parental recombinant inbred strains, which capture >90% of common genetic diversity among mouse species, used recently to study complex cardiovascular disease traits in adults (*Salimova et al., 2019*).

Here, we advance our understanding of genetic complexity underlying atrial septal variation in inbred QSi5 and 129T2/SvEms mouse strains. We used AIL to confirm and fine-map significant QTL identified for FVL and FOW in the previous F2 study. We also sequenced the genomes of the AIL parental strains and integrated variant analysis with transcriptome data from dissected atrial septa at different developmental stages in parental strains. Our results provide the first high-resolution picture of genetic complexity underpinning atrial septal variation in the mouse model, allowing the identification of QTL with high impact, candidate genes, and gene regulatory network perturbations that may have relevance to human PFO and ASD.

## Results
### Selection of AIL mice - atrial septal phenotypes

In our previous F2 study (*Kirk et al., 2006*), adult QSi5 and 129T2/SvEms parental mice and F2 mice were scored for PFO as a binary trait, and three quantitative anatomical parameters of the inter-atrial septum (FVL, FOW, and CRW) that were found to be associated with PFO (see Methods, *Figure 1C*; *Biben et al., 2000*; *Kirk et al., 2006*). The prevalence of PFO in parental strains was 4.5% and 80%, respectively. Beginning the pedigree from a single breeding pair, we randomly intercrossed F2 mice selected for extremes of phenotype for a further 12 generations to generate F14 (AIL) mice (see details in Methods). As in the original F2 study, the most relevant quantitative septal parameter in the F14 study was FVL (*Table 1*; *Figure 1—figure supplement 1*) - it showed the greatest difference

**Table 1.** Phenotypic characteristics of parental strains and F2 mice extracted from *Kirk et al., 2006* compared to F14 mice.

|  | QSi5 | 129T2/SvEms | F2 | F14 |
|---|---|---|---|---|
| N | 66 | 75 | 1437* | 933 |
| PFO (%) | 4.5 | 80 | 17 | 34 |
| FVL ± SD (mm) | 1.13 ± 0.11 | 0.60 ± 0.11 | 1.0 ± 0.19 | 1.01 ± 0.16 |
| FOW ± SD (mm) | 0.21 ± 0.06 | 0.24 ± 0.06 | 0.21 ± 0.07 | 0.24 ± 0.07 |
| CRW ± SD (mm) | 0.51 ± 0.13 | 0.44 ± 0.12 | 0.41 ± 0.12 | 0.54 ± 0.15 |
| Body Weight ± SD (g) | 29.4 ± 2.77 | 17.5 ± 2.1 | 26.6 ± 3.3 | 25.8 ± 2.9 |
| Heart Weight ± SD (g) | 0.21 ± 0.02 | 0.14 ± 0.02 | 0.21 ± 0.03 | 0.18 ± 0.03 |

*Data were incomplete for some mice.

in mean length up to a maximum of 2.5-fold between different inbred strains (*Biben et al., 2000*), and a difference of ~twofold and 4.8 standard deviations (SD) between parental strains for this study. FVL also showed the strongest (negative) correlation with PFO prevalence among a number of inbred strains (*r*=–0.97) (*Biben et al., 2000*), and in both F2 (p<0.001) and F14 (p<0.001) generations (*Kirk et al., 2006*).

In our original study in which quantitative septal parameters were defined (*Biben et al., 2000*), we also determined the width of the patent corridor between the *septum primum* and *septum secundum* in cases of PFO, measured at the edge of the *ostium secundum* of the *septum primum* remnant (*Figure 1C*), which forms a prominent crescent-shaped ridge. However, since this parameter was constrained to cases of PFO, for QTL analysis we considered only CRW (crescent width), defined as the length of the prominent crescent irrespective of the presence of PFO (*Kirk et al., 2006*). Whereas there was a strong statistical effect of PFO on CRW in the F2 study (p<0.001) (*Kirk et al., 2006*), this was lost in the F14 cohort (p=0.069) (*Supplementary file 1*). An additional observation was that mean CRW was positively associated with the risk of PFO in the F2 cohort, whereas in the F14 study longer CRW was associated with lower PFO prevalence (negative association). For these reasons and because CRW remains an ill-defined anatomical parameter, we did not consider this trait in selecting mice with extremes of phenotype in either the F2 or F14 study. However, post-hoc analysis for CRW QTL in the F2 study revealed a significant QTL on MMU7 (LOD = 4.58) and a suggestive one on MMU3 (LOD = 3.49) (*Kirk et al., 2006*). We, therefore, performed a similar post-hoc analysis in this F14 study (see below).

FOW showed more subtle differences among individuals within the parental strains, substantially overlapping within one SD (*Table 1*), and the correlation with FVL was weak in the F2 cohort (*r*=–0.087; p=0.001) (*Kirk et al., 2006*). Nonetheless, variation in FOW is reflective of up to a twofold difference in *foramen ovale* area (*Biben et al., 2000*). Therefore, FOW was taken into account in selecting mice of extreme phenotypes for inclusion in the AIL study and was indeed associated with both PFO (p<0.001) (*Supplementary file 1*) and FVL (*r*=–0.284; p<0.001) in F14 mice (*Supplementary file 2*).

In the AIL study, we sought to confirm and fine-map significant QTL found in the F2 study with LOD scores above the threshold for significance of 4.3. This included three QTLs for FVL and three for FOW. We also included a suggestive QTL (2.8<LOD<4.3) for FOW located on MMU9 (LOD = 3.43), since its peak covered the T-box transcription factor gene *Tbx20*, variants in the human orthologue of which are known to cause familial septal defects and severe PFO (*Kirk et al., 2007*).

## Selection of AIL mice - heart weight and body weight phenotypes

Given that QSi5 mice were originally selected for their high fecundity and growth, it was evident that the quantitative septal parameters under study might be influenced by heart size and mass (*Table 1*; *Kirk et al., 2006*), and indeed both FVL and FOW were significantly correlated with HW in both F2 and F14 cohorts, albeit that the effects were small (*Supplementary file 2*; *Kirk et al., 2006*) and HW had no influence on the likelihood of PFO. In the F2 study, therefore, we did not normalize septal data for HW so as not to mask important QTL (*Kirk et al., 2006*). However, the possibility that FVL and FOW QTL could be explained by variation in HW or BW has not been formally excluded. Thus, prior to the analysis of AIL mice, we performed a retrospective linkage analysis for HW and BW on F2 data. The HW of F2 mice was initially adjusted for factors with significant effects (age, sex, and BW) and we used the same LOD score criteria as in the F2 study (4.3 for significant and 2.8 for suggestive linkage) (*Lander and Kruglyak, 1995*). We discovered a suggestive QTL (LOD = 3.4) for normalized HW on MMU7, and another on the same chromosome that fell just short of suggestive (*Figure 2A*). The suggestive HW QTL overlapped with previously determined QTL for BW on MMU7, however, did not represent a BW QTL in our study. We also found a significant QTL for HW normalized for age, sex and BW on MMU11 (LOD = 8.5) (*Figure 2B*). This QTL overlapped one for BW normalized for age and sex (LOD = 14.2) (data not shown). Thus, we included normalized HW as a parameter in selection of F14 mice with extremes of phenotype for further QTL analysis (see Methods), and selected markers for fine mapping of the significant normalized HW QTL on MMU11.

## Linkage results for atrial septal morphology

We set a LOD score of 2 as a cut-off for significant linkage based on the density of chosen markers (~2 cM) and the size of the genomic regions covered (*Lander and Botstein, 1989*). From the septal

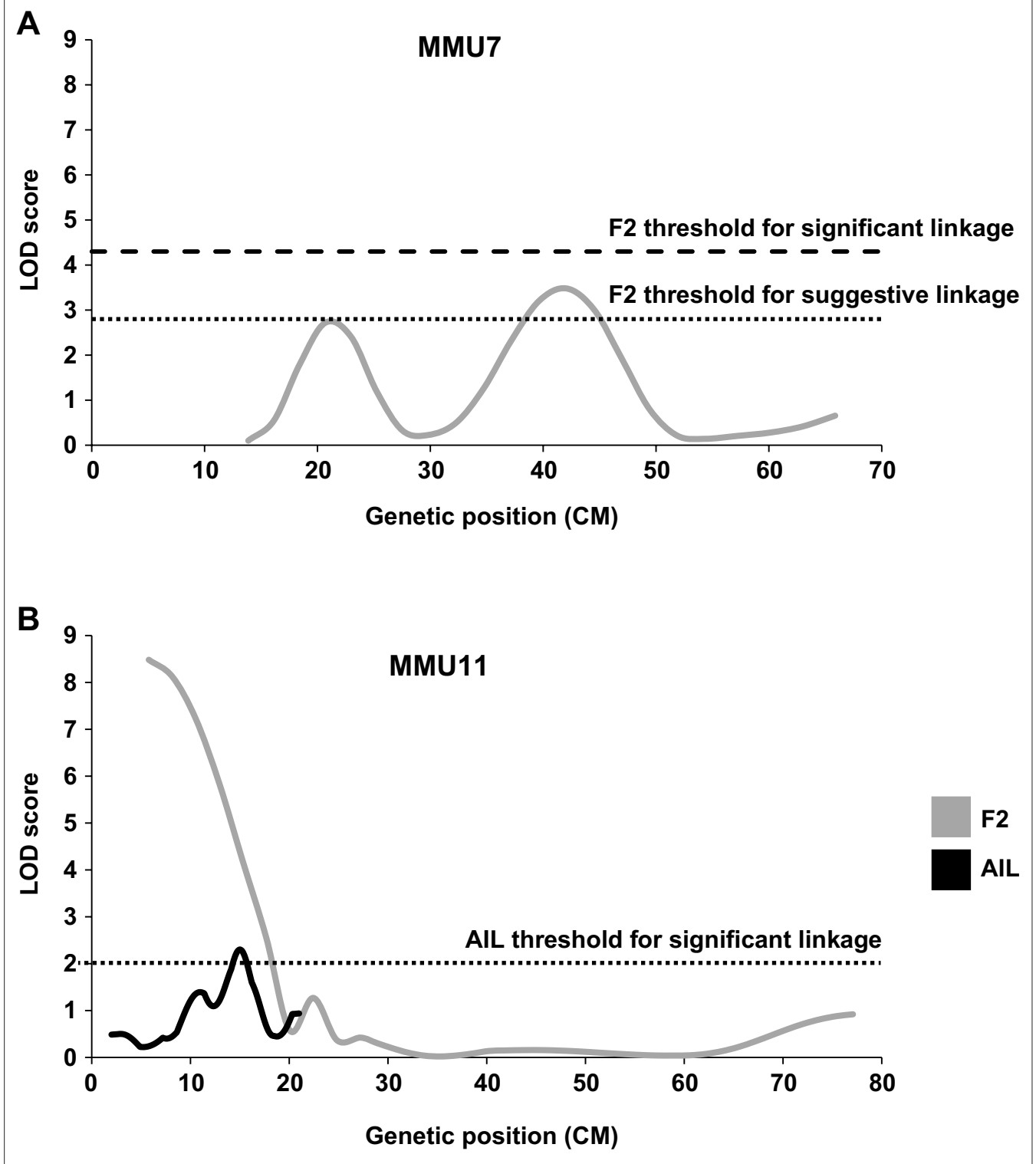

**Figure 2.** Linkage analysis for heart weight (HW). Suggestive and significant quantitative trait loci (QTL) identified on MMU7 (**A**) and MMU11 (**B**) by retrospective analysis of HW in the F2 study, the latter being fine-mapped by the advance intercross line (AIL) study. Y-axes represent logarithm of odds (LOD) scores for HW adjusted for age, sex, and body weight (BW).

The online version of this article includes the following figure supplement(s) for figure 2:

**Figure supplement 1.** Comparison of advanced intercross line (AIL) results for heart weight (HW) (red line) and quantitative traits of atrial septum.

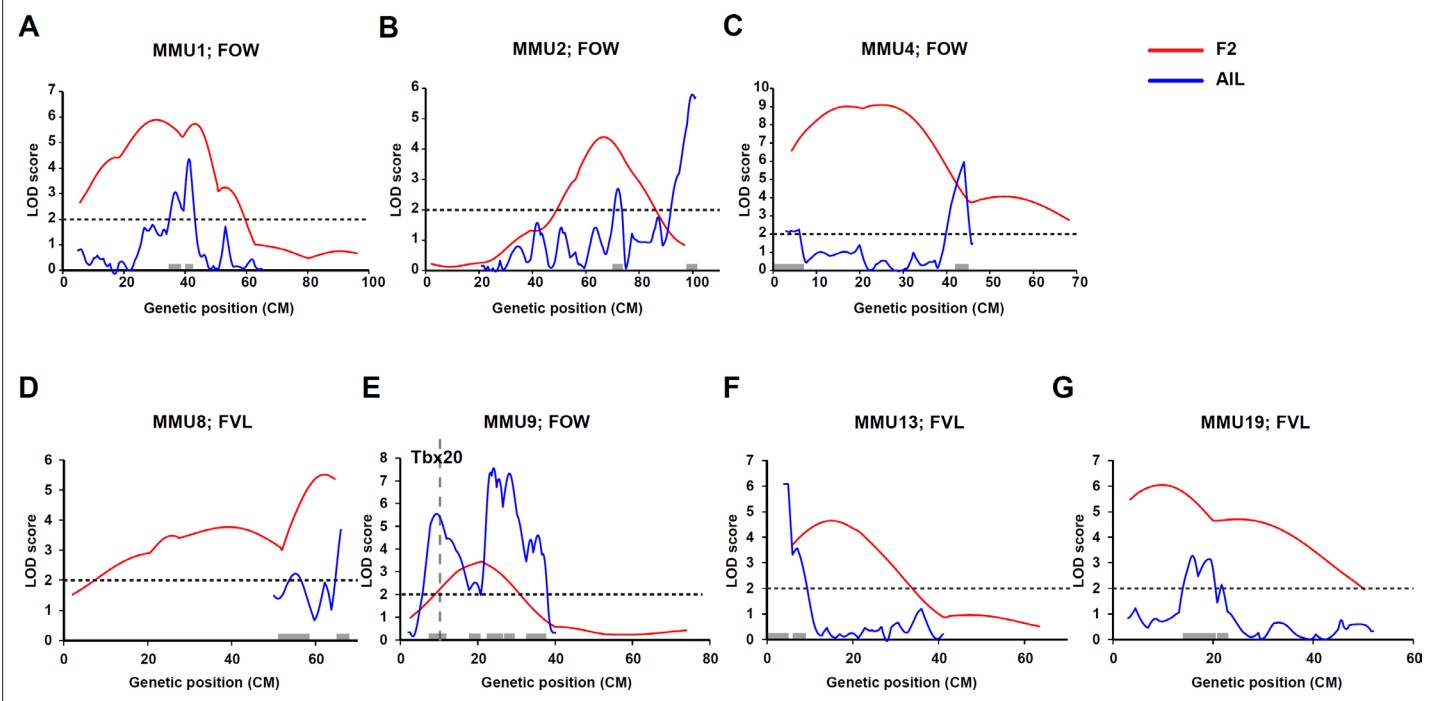

**Figure 3.** Comparison of linkage results from F2 (red line) and advanced intercross line (AIL) (blue line) populations. Y-axis represent logarithm of odds (LOD) scores and x-axis represent genetic map positions. A 1-LOD drop-off for each quantitative trait locus (QTL) is shown on the x-axis representing the confidence interval of the QTL.

The online version of this article includes the following figure supplement(s) for figure 3:

**Figure supplement 1.** Differentiation of overlapping QTL peaks identified by the advanced intercross line (AIL) data.

morphology QTL identified in the F2 study (six significant and one suggestive), at least six QTLs were confirmed and significantly narrowed using AIL data. Five F2 QTLs resolved into multiple peaks and several new QTLs were also discovered. Furthermore, the overlap between QTL for different traits was increased. *Supplementary file 3* describes QTL for FVL, FOW, and CRW identified by the AIL study, and *Figure 3* shows the examples from each relevant chromosome comparing F2 and AIL mapping results. Overall, 37 QTLs were significant in the AIL study. As in the F2 study, the direction of QTL varied, and were classified here as 'normal' if they contribute to the negative septal features documented for the 129T2/SvEms strain (shorter FVL; longer FOW; shorter CRW) and 'cryptic' if protective against septal defects (i.e. contributing to the QSi5 phenotype of longer FVL; shorter FOW; longer CRW) (*Supplementary file 3*).

## Linkage analysis of PFO as a binary trait

Although in general, the linkage analysis of continuously distributed traits is more sensitive and informative than the analysis of binary traits, the analysis of PFO as a binary trait was still of importance in our study. Therefore, we also performed direct linkage analysis for the presence or absence of PFO in the F14 mice as a binary trait using a logistic regression model that we previously developed (*Moradi Marjaneh et al., 2012*). This analysis confirmed most of the FVL and FOW QTL. The AIL results for each chromosome for quantitative (FVL, FOW, and CRW) and binary (PFO) atrial septal traits are compared in *Figure 4*. Results from each chromosome are summarized as follows:

### MMU1

We previously identified a significant QTL for FOW with a peak at 30.8 cM extending 26.1 cM (inclusive of 1-LOD on each side of the peak = 1-LOD drop-off) on MMU1 (*Figure 3A*). The linkage analysis of AIL data narrowed this region to 7.5 cM, including two adjacent peaks with LOD scores of 3.1 and 4.4 overlapping in the 1-LOD drop-off. To assess whether the two peaks were distinct, we re-ran the linkage analysis including the marker closest to the distal peak as a fixed term (*Figure 3—figure*

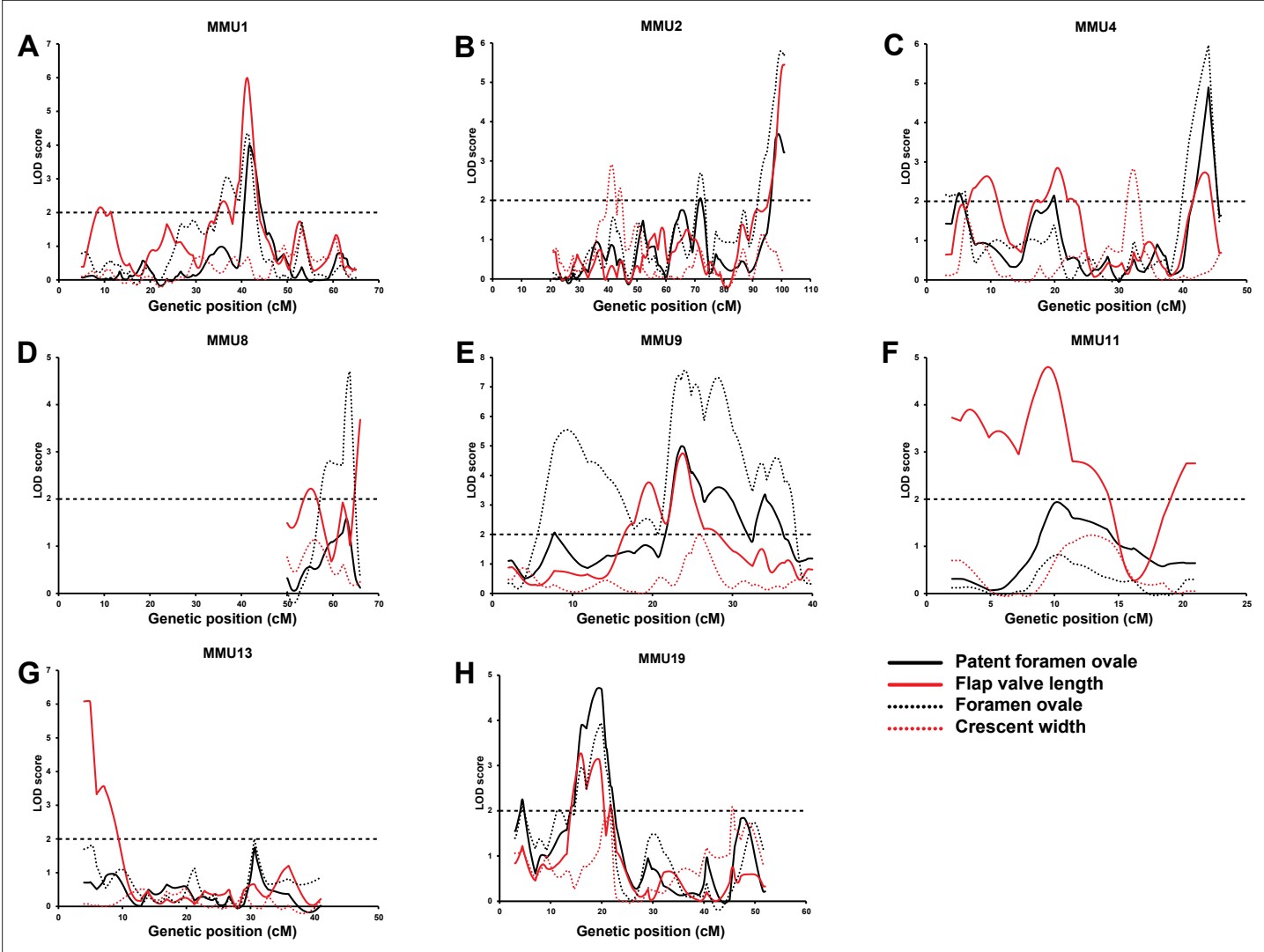

**Figure 4.** Comparison of advanced intercross line (AIL) results for patent *foramen ovale* (PFO), flap valve length (FVL), *foramen ovale* (FOW), and crescent width (CRW).

supplement 1A). While this led to a significant reduction in the LOD score of the distal peak, the proximal peak did not change and was still significant, indicating that the peaks represent two independent QTLs. Unlike the F2 study, the AIL study also showed strong evidence of linkage across this region for FVL (including two FVL QTL with LOD scores of 6.0 and 2.3, respectively) (*Figure 4A*). Analysis of the AIL also revealed an additional QTL for FVL on MMU1 located at 9 cM, with a peak LOD of 2.2, which is clearly separate from the original QTL (*Figure 4A*). The QTL map from the analysis of PFO as a binary trait followed the pattern produced by FOW and FVL data (*Figure 4A*). In particular, its peak at 41.6 cM (LOD = 4) strongly supported highly significant FOW and FVL QTL in that vicinity. The 1-LOD drop-off of this peak covered approximately 2.4 cM, a very substantial refinement on the F2 study.

## MMU2

The broad F2 QTL peak for FOW (peak at 67.5 cM; *Figure 3B*) was narrowed to a sharp peak at 72 cM refining the QTL genomic region from 18.2 cM (F2) to 3.6 cM (AIL) (*Figure 3B* and *Figure 4B*). We also observed a new highly significant QTL at the distal end of this chromosome (~100 cM) affecting both FOW and FVL (LOD scores 5.79 and 5.44, respectively) (*Figure 3B* and *Figure 4B*). The binary analysis of PFO supported QTL for both FOW and FVL. While the F2 study was not able to detect

QTL for CRW on this chromosome, linkage analysis of the AIL data revealed two possibly distinct QTL close to each other at 41.25 cM and 44 cM (*Figure 4B* and *Figure 3—figure supplement 1B*) without pleiotropic effects on the other quantitative traits.

### MMU4

The very broad F2 FOW QTL spanning 23.8 cM of the chromosome (peaking at 25.4 cM; *Figure 3C*) resolved in the AIL as two FOW QTL (6 cM, LOD = 2.2; 44 cM, LOD = 6.0), leaving the mid portion including the 1-LOD drop-off of the original peak unlinked (*Figure 3C*). Given the sparsity of markers across this region in the F2 study, this outcome likely represents a resolution of the broad F2 QTL region into two widely spaced QTLs. The AIL data also revealed three QTLs for FVL, of which the distal one was located at the same position as the distal FOW QTL (*Figure 4C*). The analysis of PFO as a binary trait generated a similar pattern to the FVL and FOW results. A new QTL for CRW was detected at 32.25 cM which, as for the CRW QTL on MMU2, did not affect the other traits.

### MMU8

The region at the end of this chromosome was found to be linked to FVL in the F2 study (*Figure 3D*). It peaked at 62.5 cM and covered 12.2 cM of the chromosome. On analysis of the AIL data, this QTL resolved into two QTLs at 52.25 cM and 66 cM (*Figure 3D*). A highly significant QTL was also detected for FOW at 63.7 cM (*Figure 4D*).

### MMU9

The F2 study revealed a suggestive QTL for FOW peaking at 20.7 cM (LOD = 3.4) and extending 17.3 cM on MMU9 (*Figure 3E*). The AIL resolved this into at least four separate QTLs with peaks at 9.25, 19.25, 24, and 28.25 cM (*Figure 3E*). *Tbx20*, a cardiac transcription factor gene, is located at 10.25 cM, within the 1-LOD drop-off of the first QTL, and very close to its peak at 9.25 cM. We also detected a new QTL for FOW, peaking at 35.5 cM with a maximum LOD score of 4.6. Analysis of the AIL also identified two peaks for FVL at 19.5 cM and 23.75 cM with significant LOD scores (*Figure 4E*), both overlapping with significant peaks for FOW. The analysis of PFO as a binary trait resulted in a strikingly similar pattern to the FOW, significantly supporting all four of the FOW QTL, including the QTL coinciding with the *Tbx20* gene.

### MMU11

The F2 study did not show a linkage of any region of MMU11 to the atrial septal traits. However, this chromosome was included in the AIL study to fine-map the HW QTL detected by the analysis of the F2 data (see below). Using the chosen markers for the HW QTL we also discovered at least two new QTL underlying FVL, with a peak LOD score of 4.8 at 9.5 cM (*Figure 4F*).

### MMU13

The AIL narrowed down the broad genomic region of the F2 QTL for FVL (*Figure 3F*) from 19.4 cM to 8.7 cM (*Figure 4G*). It is notable that the AIL data resulted in a shifting of the QTL peak from 15.3 cM (F2) toward the telomere of the chromosome including two close peaks at 4.9 and 7 cM (AIL) which were determined to be significantly distinct using the fixing method (see Methods) (*Figure 3—figure supplement 1C*).

### MMU19

The broad F2 QTL for FVL at 10.2 cM resolved into three separate QTLs with the highest LOD score at 16 cM (*Figure 3G*). The fixing method showed these peaks represented three separate QTLs (*Figure 3—figure supplement 1D*). We also observed a linkage between this region and FOW, with the highest LOD score of 3.9 at 19.75 cM (*Figure 4H*). Binary analysis of PFO showed a similar pattern to that seen for FVL and FOW, strongly supporting the QTL located between 13 cM and 23 cM.

## Linkage analysis for heart weight

As noted above, our retrospective analysis of F2 data revealed a significant QTL for normalized HW at the proximal end of MMU11, with the highest LOD score of 8.5, and a suggestive QTL on MMU7

peaking at 42.1 cM (*Figure 2*). This is the first report of QTL affecting HW on MMU11, although numerous HW QTL on other chromosomes has been detected previously using different cohorts and study designs (*Reed et al., 2008*; *Rocha et al., 2004*; *Sugiyama et al., 2002*). An extra nine markers covering the MMU11 HW QTL region were selected for this AIL study (see Methods). We also performed linkage analysis for normalized HW on other chromosomal regions for which markers had been previously chosen for analysis of atrial septal morphology. MMU2, MMU4, MMU9, MMU11, and MMU13 all showed significant evidence of linkage to HW *Figure 2—figure supplement 1*; of these, only MMU2 has previously been linked with HW, albeit on different genetic backgrounds (*Rocha et al., 2004*). On MMU11, we observed evidence of linkage distally with a LOD score of 2.3 peaking at 15 cM, thus confirming the original F2 QTL (*Figure 2B*).

Comparison of linkage results for HW and atrial septal morphology, however, showed that only the HW QTL on MMU2 and MMU9 had a potential overlap within its 1-LOD drop-off with QTL for atrial septal parameters. Thus, the chromosomal locations of QTL for HW were largely different from those defining atrial septal parameters.

## Contribution of QTL to phenotypic differences

The percent attributable phenotypic variance for each QTL is shown in *Supplementary file 3*. As in the F2 study (*Kirk et al., 2006*), many AIL QTL were of relatively large effect - 18/37 individually contributed ~23–70% of the difference between parental means. Interestingly, the majority of these (15/18) were for FOW, with seven being normal and eight being cryptic QTL. Normal and cryptic QTL are anticipated to interact additively or in more complex ways in parental strains and individual mice, which may show phenotypes more extreme than parental strains (*Kirk et al., 2006*). The effect sizes of FVL QTL were somewhat smaller, individually 0.3–9% of the difference between parental means, with the majority (12/15) being normal QTL. Whereas it is difficult to precisely quantify the effects of individuals or combinations of QTL, our data suggest that many of the QTL detected are of large or moderate effect size and, therefore, collectively contribute significantly to quantitative trait variation between parent strains.

## Whole genome sequencing of parent strains and SNP density analysis

The whole genomes of the AIL parental strains were sequenced and mapped to the C57Bl/6J mouse reference genome (NCBI38/mm10). Genomic analysis of the various inbred mouse strains used in research reveals a mosaic pattern of haplotypes attributable to the genome of founder lines. Accordingly, pairwise comparisons of inbred strain genomes show long blocks of either low or high SNP density representing shared and divergent ancestry, respectively (*Frazer et al., 2007*; *Wade et al., 2002*). Regions with a high rate of polymorphism between parental strains (spanning nearly one-third of the genome but harboring more than 95% of the genetic variation) can significantly reduce the regions of interest underlying QTL (*Wade et al., 2002*), albeit that regions with low or high polymorphism may also carry variants that have arisen since the establishment of parent strains (*Bloom et al., 2019*). Although these would be rare, they could in, principle, also contribute to septal variation.

High-quality homozygote variants between the AIL parental strains identified by whole genome sequencing were used to compute variant density across the genome (*Figure 5—figure supplement 1*; see Methods). To assess the distribution of variant densities, we binned the genome into non-overlapping intervals and counted the number of variants in each bin (*Figure 5—figure supplement 2*). The analysis was repeated using different bin sizes ranging from 50 Kb to 1.1 Mb. The distribution of variant densities across the genome showed a bimodal pattern, confirming two categories of genomic regions with low and high variant density regions. Binning at 600 Kb intervals showed the clearest distinction between the two peaks of distribution and the boundary between the two peaks, representing a cut-off of 1000 variants per 600 Kb interval, was chosen to classify genomic segments as low or high variant density regions. Intersecting with high variant density regions narrowed the total size of the QTL regions from 264.1Mb to 109.5 Mb (41.5%) (*Figure 5*). It is noteworthy that among the high variant density regions called there were blocks of very high variant density, presumably also derivative of strain ancestry.

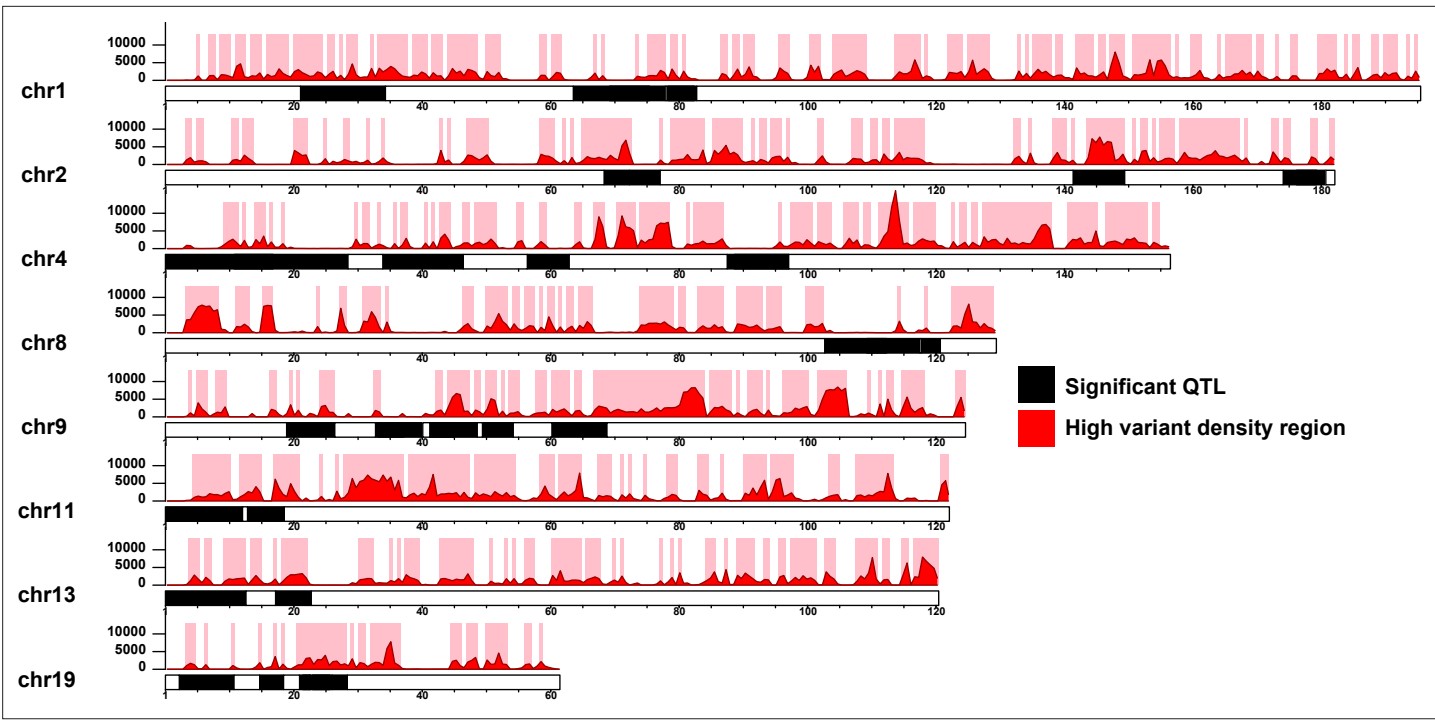

**Figure 5.** Segmentation of the quantitative trait loci (QTL) according to high variant density regions. The QTL regions for each chromosome are illustrated as black blocks. The red histograms and y-axeis represent the variant density defined as the number of 129T vs QSi5 variants per 600 Kb genomics intervals. High variant density regions (density ≥ 1000) are represented as pink blocks. The x-axis for each chromosome represents the genomic position (Mb).

The online version of this article includes the following figure supplement(s) for figure 5:

**Figure supplement 1.** Combined genome-wide variant zygosity of advanced intercross line (AIL) parental strains.

**Figure supplement 2.** Assessment of 129T2/SvEms vs QSi5 variant density across the genome using different bin sizes.

## Filtering for high-impact variants and candidate genes

There were a total of 6,769,034 high-quality variants between the AIL parental strains identified by whole genome sequencing (using C57Bl/6J as reference). We first filtered for protein-coding variants in genes expressed in the developing atrial septum (see below) and which lay under QTL regions (within the 1-LOD drop-off support interval but without filtering for high SNP density regions), then analyzed through the Ensembl Variant Effect Predictor (VEP) tool (*McLaren et al., 2016*) for pathogenicity impact. We filtered variants for high impact (stop-gain, stop-loss, start-loss, splicing defect, frameshift), missense variants predicted deleterious by SIFT (*Ng and Henikoff, 2003*), and non-high impact (e.g. non-frameshift) deletions. This resulted in 45 variants spread over 28 genes for 129T2/SvEms and 47 variants across 35 genes for QSi5 (*Supplementary file 4*; *Figure 6A and B*), noting that two genes had unique variants in each parent strain (total number of genes carrying predicted deleterious variants = 61). Virtually all of these were in high SNP density regions (84/92; 91.3%; *Supplementary file 4*) and many genes (26%) contained multiple variants (*Figure 6A and B*).

For the *Tbx20* gene on MMU9, a total of 321 variants were detected (*Supplementary file 5*); however, only three were exonic and these were synonymous. No defects in splice motifs or cryptic splice donor/acceptor sites were detected using the VEP tool. We reran the analysis using Spliceogen (*Monger et al., 2019*) and again no splicing defects were detected. Spliceogen provides rankings of possible cryptic splice sites (*Supplementary file 5*), however, these do not reflect actual probabilities. Therefore, we analyzed splice junctions detected in RNA-seq data from the developing septum (see below) but did not find splice site differences for *Tbx20* between parental strains. We also mapped *Tbx20* transcript isoforms with Salmon (*Patro et al., 2017*) and performed differential expression analysis, again yielding no differences in isoform expression between strains (*Supplementary file 5*).

To create a high-confidence QTL gene list, we annotated genes lying under QTL according to prior association with cardiovascular phenotypes (Mouse Genome Informatics; MGI) or link to human

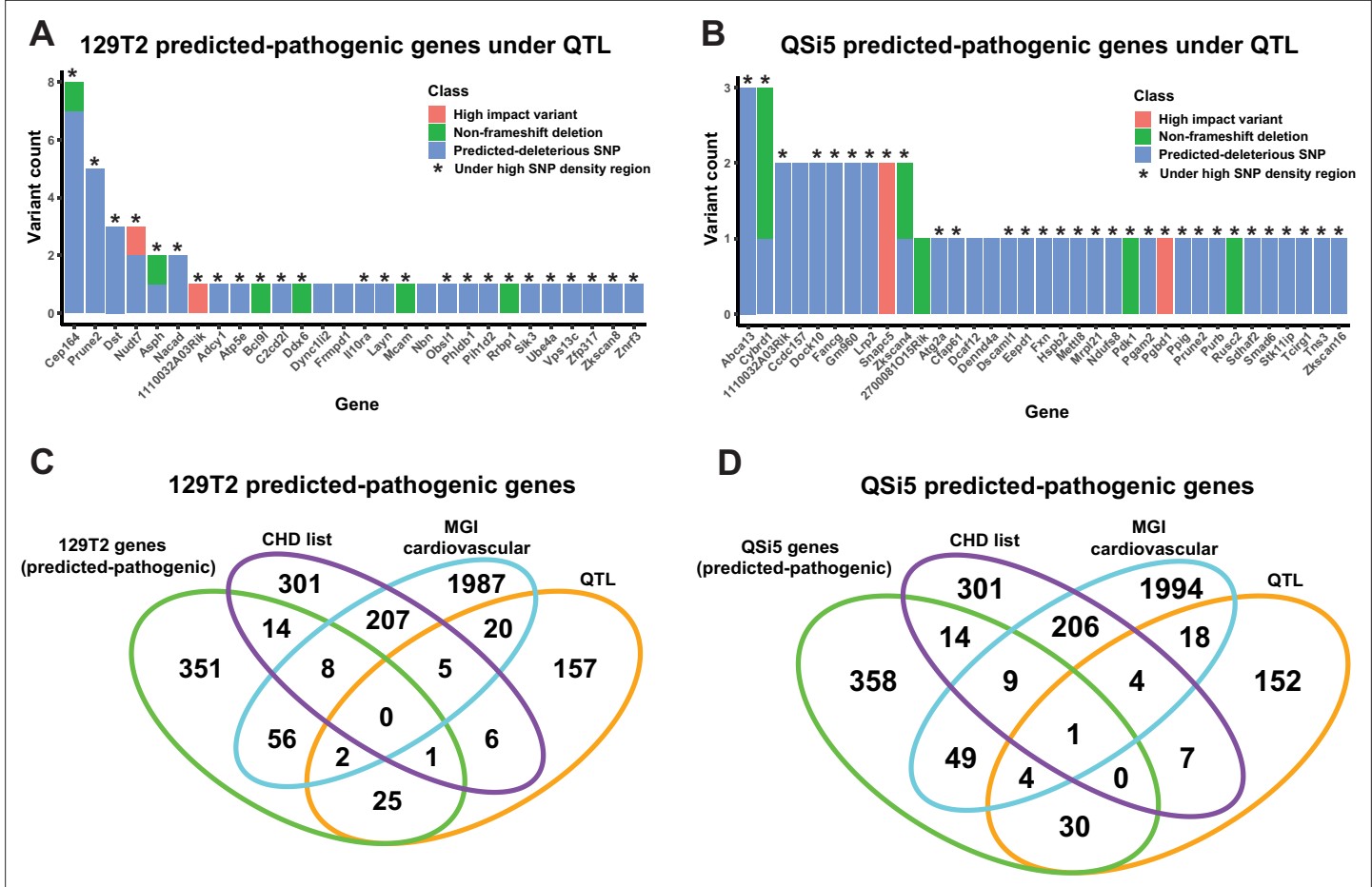

**Figure 6.** Identification of predicted-pathogenic variants. (**A, B**) Number of variants per gene for predicted pathogenic genes under quantitative trait loci (QTL) according to classification (high-impact, non-frameshift deletion, or predicted-deleterious missense variant) for (**A**) 129T2 and (**B**) QSi5. (**C, D**) Overlap of genes containing predicted-pathogenic variants with (1) genes in a high confidence/emerging congenital heart disease (CHD) list, (2) genes with a heart development cardiovascular function from mouse genome informatics (MGI), and (3) all genes under QTL intervals that contain protein-modifying variants. Overlaps are shown for predicted pathogenic variants in (**C**) 129T2 and (**D**) QSi5.

CHD from a curated list of high-confidence and emerging CHD genes (*Alankarage et al., 2019*). Eight genes (*Cybrd1, Dst, Fxn, Lrp2, Mcam, Pgbd1, Sik3, Smad6*) intersected with one or both lists (*Figure 6C and D*). We highlight below several high-confidence genes, as well as *Cep164*, an example of a gene in which seven predicted-deleterious variants were detected:

## Cep164

129T2/SvEms strain; MMU9; under a cryptic QTL for FOW (effect size 65%). *Cep164* contained seven predicted-deleterious missense coding variants and one in-frame three base pair (bp) deletion (*Figure 6A*). *Cep164* encodes a docking protein associated with mother centrioles and is essential for the formation of primary and multi-cilia, therefore, affecting cellular processes such as fluid sensing and inter-cellular signaling (*Schmidt et al., 2012*). Defects in ciliogenesis are well known to cause CHD (*Audain et al., 2021*; *Li et al., 2015*). Three *Cep164* variants overlap regions predicted to encode coiled-coil domains in the centre of the protein and one variant may overlap a predicted DNA/RNA-binding domain (PredictProtein *Bernhofer et al., 2021*). *Cep164* knockout mice are embryonic lethal showing multiple severe defects including heart developmental arrest at the looping stage (*Siller et al., 2017*).

## Dst

129T2/SvEms strain; MMU1; under a cryptic QTL for FVL (effect size 5.68%). *Dst1* carried three missense mutations predicted to be deleterious. *Dst1* encodes diverse isoforms of Dystonin, members

of the plakin family of cytoskeletal docking and tumor suppressor proteins implicated in cell growth, adhesion, cytoskeleton organization, intracellular transport, and migration. They may function in part through an ability to antagonize the activation of the Hippo pathway effector YAP (*Jain et al., 2019*). Mutations in humans and mice cause fragile skin (epidermolysis bullosa), severe neurological conditions, as well as skeletal muscle instability (*Horie et al., 2017*; *Künzli et al., 2016*). Isoform BPAG1b is expressed in cardiac and skeletal muscles colocalizing with Z-bands, sarcolemma, and cardiac intercalated discs, and interacts with α-actinin2. Knockout strains show diminished heart size and atrophy before weaning and altered hypertrophy/wall stress markers (*Horie et al., 2017*). *Dst* is classified as an emerging CHD gene.

## Sik3

129T2/SvEms strain; MMU9; under a cryptic QTL for FOW (effect size 65%). *Sik3* carries a nonsynonymous SNP predicted to affect SIK3 function. *Sik3* encodes a serine/threonine kinase of the AMP Kinase (AMPK) family whose activity is inhibited by cyclic AMP (cAMP) via phosphorylation at multiple sites by a cAMP kinase (PKA). SIK3 functions in a variety of biological processes (*Wein et al., 2018*) including through phosphorylation and inhibition of cAMP-regulated transcriptional coactivators (CRTCs) and class IIa histone deacetylases (HDACs), the latter functioning as inhibitors of MEF2 transcription factors that play key roles in heart development (*Darling and Cohen, 2021*). *Sik3* overlaps with the MGI list with a homozygous targeted EUCOMM mutation showing increased heart weight.

## Lrp2

QSi5 strain; MMU2; under a normal QTL for CRW (effect size 30%). *Lrp2*, encoding a low-density lipoprotein receptor-related protein, carried two missense variants predicted to be deleterious. *Lrp2* is expressed in the SHF progenitor pool and developing outflow tract (OFT) and is known to act in multiple morphogenetic pathways. Variants in human *LRP2* have been associated with hypoplastic left heart (*Theis et al., 2020*), whereas mutant mice show premature differentiation and depletion of anterior SHF progenitors in the fetus, leading to a shortened OFT, *truncus arteriosus*, and ventricular wall and septal defects (*Christ et al., 2020*).

## Smad6

QSi5 strain; MMU9; under a cryptic QTL for FOW (effect size 58%). *Smad6*, encoding an inhibitory member of the *mothers against decapentaplegic* transcription factor (SMAD) family, carried the nonsynonymous SNP NM_008542 c.845C>G (p.Arg282Pro), predicted to be deleterious. We chose this variant for further analysis. SMAD6 is an inhibitor of BMP signaling acting at multiple levels, including inhibition of the binding of receptor-activated (R) SMADs 1, 5, and 8 (the effector transcription factors of the BMP pathway) to BMP receptors, where they are normally activated by phosphorylation; inhibition of binding of R-SMADs to their co-SMAD (SMAD4); and direct transcriptional repression of targets genes of R-SMADs by recruiting transcriptional repressors such as CtBP, HDAC-1/3 and HOXC8/9 (*Bai and Cao, 2002*; *Bai et al., 2000*; *Lin et al., 2003*; *Figure 7A*). BMP signaling has been implicated in the cardiac specification and development of cardiac chambers, valves, septa, and outflow tract (*Wang et al., 2011*), and SMAD6 mutations in humans are associated with syndromic and non-syndromic congenital malformations, including cardiac outflow tract defects, as well as patent *foramen ovale* and atrial and ventricular septal defects (*Calpena et al., 2020*; *Galvin et al., 2000*; *Tan et al., 2012*). The R282P variant lies within the regulatory linker region joining the MH1 DNA-binding domain and MH2 transactivation/protein:protein interaction domain (*Figure 7B*), and interrupts the conserved PPXY motif known to bind the E3 ubiquitin ligase SMURF1, predicted to poly-ubiquitinate SMAD6 triggering proteosome-mediated degradation (*Sangadala et al., 2007*). SMURF1 ubiquitinates other targets including SMAD1, and can also act to exclude SMAD1 (and possibly SMAD6) from the nucleus (*Sapkota et al., 2007*). The R282P variant also lies close or immediately adjacent to serine residues that are modifiable by phosphorylation (*Bian et al., 2014*; *Christensen et al., 2010*; *Mertins et al., 2016*). In SMAD1, SMURF1 binding and subsequent ubiquitination is dependent upon MAPK-mediated phosphorylation of linker serine/threonine residues, which prime phosphorylation by kinase GSK-3β, facilitating poly-ubiquitination (*Sapkota et al., 2007*). Thus, the R282P SMAD6 variant potentially disrupts the complex regulatory network regulating SMAD6 stability and activity.

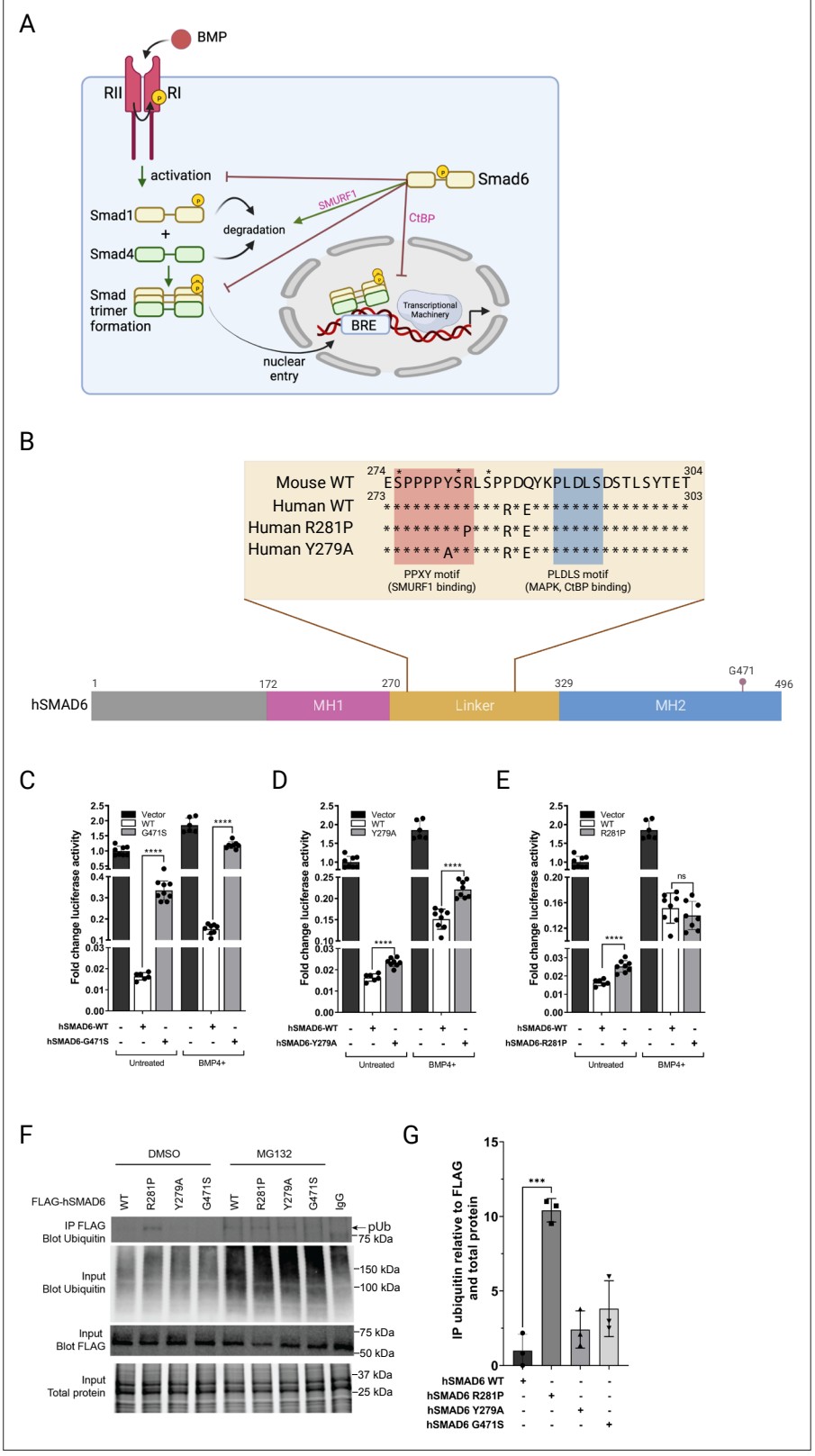

**Figure 7.** Functional analysis of SMAD6 variant R282P. (**A**) Schematic of BMP signaling pathway highlighting roles of SMAD6. Prepared with *Biorender*. (**B**) Schematic of hSMAD6 showing the three main functional domains; the conserved DNA-binding MH1 domain (172-270aa), regulatory linker region (271-328aa), and the C-terminal transactivation/cofactor interaction MH2 domain (329-496aa). The position of G471 within the MH2 domain is

*Figure 7 continued*

indicated – mutation of this residue (G471S). Insert shows the amino acid sequence of the linker region containing PPXY (SMURF1 binding) and PLDLS (CtBP binding) domains, with the position of hSMAD R281P and Y279A variants indicated. Known serine phosphorylation sites are marked * above the sequence. (**C–E**). Functional analysis of hSMAD6 variants. Activity of hSMAD6-G471S (**C**), hSMAD6-Y279A (**D**), and hSMAD6-R281P (**E**), were assessed relative to wild-type (WT) hSMAD6 on a BMP-signaling responsive (BRE)-luc promoter in untreated and BMP4-treated cells. n=6–9; differences between WT and variant were assessed by unpaired t-test; data presented as mean ± SD. Asterisks denote significance, ****p<0.0001, ns = 0.3332. (**F,G**) Ubiquitination of WT hSMAD6 and variants assessed in MG132 or DMSO (control)-treated HEK293T cells. WT FLAG-ShMAD6 and variants were immunoprecipitated with anti-FLAG antibody with lysates subject to western blotting with anti-ubiquitin antibody (**F**). Total ubiquitin, FLAG-hSMAD6, and protein indicated. (**G**) Quantification of FLAG-hSMAD6 ubiquitination. n=3; difference between WT and variants assessed by one-way ANOVA with Tukey's multiple comparisons; data presented as mean ± SD. Asterisks denote significance, ***p=0.0001.

The online version of this article includes the following source data for figure 7:

**Source data 1.** Original blots associated with *Figure 7*.

We chose to follow up on this SMAD6 variant. First, we analyzed the activity of WT and mutant forms of SMAD6 in HEK293T cells after transfection of human (h) cDNAs and a luciferase reporter construct reading out SMAD-dependent BMP-signaling (*Figure 7C–E*). WT hSMAD6 repressed BMP signaling by ~98% at baseline and ~92% after stimulation of cells with BMP4. Repression was diminished significantly by the hSMAD6 mutation G471S, which abolishes the interaction between SMAD6 and SMAD1 (*Figure 7C*). Then, to gauge the impact of the loss of interaction with SMURF1, we replaced tyrosine 279 of the hSMAD6 PPXY domain, predicted to be essential for SMURF1 binding, with alanine (*Sangadala et al., 2007*). Relative to WT, the Y279A variant slightly reduced SMAD6 repression at baseline, and the change was greater after BMP4 stimulation (*Figure 7D*). The R281P hSMAD6 variant (equivalent to R282P in mouse), also showed a slight reduction in repression similar to Y278A at baseline, however, the reduction was eliminated after BMP4 stimulation (*Figure 7E*), suggesting an alternative mechanism to simply loss of SMURF1 binding. To take a different approach, we tested the poly-ubiquitination status of the above hSMAD6 variants using co-immunoprecipitation (*Figure 7F*). After transfection of FLAG-tagged hSMAD6 vectors into HEK293T cells and treatment with proteosome inhibitor MG132 or DMSO control, lysates were precipitated using a FLAG antibody and ubiquitination of hSMAD6 detected by western blotting. In the presence of MG132, poly-ubiquitinated hSMAD6 WT, R281P, and Y279A were readily detected (*Figure 7F*). Ubiquitination of G471S was lower, perhaps because ubiquitination depends on interaction with an R-SMAD. In DMSO, poly-ubiquitination of WT, Y279A, and G471S hSMAD6 were barely detected; however, poly-ubiquitination of R281P occurred at levels ~10 fold higher than WT, comparable to that seen in the presence of MG132 (*Figure 7F and G*). Our findings suggest a defect in the hSMAD6 R281P-SMURF1 binding/ubiquitination pathway leading to constitutive poly-ubiquitination.

Because the *Smad6* variant lies under a cryptic QTL of high effect (58%) and is seen in the QSi5 strain (with robust septal qualities), it is an excellent candidate for being protective against atrial septal defects. We confirmed *Smad6* expression in the developing interatrial septum (see below). Other highlighted variants have not yet been validated.

## Transcriptome analysis of the developing atrial septum

To explore changes in the gene regulatory networks affecting atrial septal traits, we profiled the transcriptome of developing septa dissected from AIL parental strains at embryonic stages (E) 12.5, E14.5, and E16.5 (see Methods). At the earliest time point, E12.5, the thin septum primum has already fused with the AV septum via its mesenchymal cap, and the ostium secundum has been created by cell death; thus, transcriptome changes across the three stages analyzed may reflect prior changes during septum primum formation, as well as the formation of the septum secundum, septal growth, and remodeling. RNA was extracted and pooled into two biological replicates per mouse strain per time point (three septa/pool; total 12 samples) and cDNA libraries were generated and sequenced on an Illumina platform.

Principal component analysis (PCA) performed on the top 500 variable protein-coding genes showed that sample replicates clustered closely together, whereas samples from the different embryonic time

points segregated along the first principal component (PC1) axis and those from the different inbred strains segregated along the PC2 axis (*Figure 8A*). We calculated differentially expressed protein-coding genes (DEGs; protein coding) for each time point separately (using the *DESeq2* R package *Love et al., 2014*) using significance thresholds of $p_{adj}$ <0.05 and log2 fold-change >0.5. E12.5 had the largest number of differentially expressed genes (990), followed by E16.5 (881) and E14.5 (762) (*Figure 8B and C*; *Supplementary file 6*), and at all time points there were more genes downregulated in 129T2/SvEms compared to QSi5 (*Figure 8C*). The majority of DEGs were unique to a time point; however, 453 genes were downregulated in at least 2-time points. Almost 30% (18/61) of genes with predicted-pathogenic variants showed differential expression at at least 1-time point (*Supplementary file 4*).

We investigated the biological significance of DEGs at each time point separately using protein-protein association connections obtained from the *STRING* database (*Szklarczyk et al., 2017*), filtering for high-confidence connections. Overall network significance was calculated by comparing network metrics with those from *STRING* networks of randomly selected genes across 100,000 permutations. To discover potential drivers of network perturbations, we also included in the *STRING* analysis protein-coding DEGs which were located under QTL (1-LOD drop-off) and in which variants predicted to be deleterious had been identified. Using this stringent approach, the network at E14.5 was significant for both the number of edges (observed = 921; expected = 419.8 ± 56.5; p<1 × $10^{-5}$) and average clustering coefficient (observed = 0.40; expected = 0.27 ± 0.03; p=8 × $10^{-5}$) (*Figure 8D*). The E14.5 network was driven predominantly by genes downregulated in 129T2/SvEms compared to QSi5 septa, i.e., the strain with the highest prevalence of PFO and most disadvantageous septal traits (shortest FVL, largest FOW). Highly connected sub-networks contained genes involved in biosynthetic and signaling pathways and macromolecular machines, including nucleosomes, ribosomes, mitochondria, and extracellular matrix (ECM) (*Figure 8D*). This signature was supported by a significant over-representation of gene ontology (GO) biological process terms among DEGs for *nucleosome assembly, chromatin assembly, translation, ATP metabolic process, and extracellular matrix organization* (*Supplementary file 7*; Fisher's exact test; false discovery rate <0.05). Whereas networks for E12.5 and E16.5 alone, and those for DEGs common across all time points, were not significant overall, both E12.5 and E16.5 networks contained ribosomal and ECM protein sub-networks (*Figure 8—figure supplement 1* and *Figure 8—figure supplement 2*).

Within the E14.5 network, Ubiquitin C (*Ubc*) was a prominent downregulated hub gene connecting directly to histones (specifically H2B) and ribosomal genes, as well as others, dispersed across the network, most of which were also downregulated (*Figure 8D*). In addition to its well-characterized role in protein homeostasis, *Ubc* is also involved in transcription and genome integrity (*Mark and Rape, 2021*; *Mattiroli and Penengo, 2021*; *Qu et al., 2021*). As such it might be expected to be functionally connected to a large number of genes even in random networks; however, the number of connections between *Ubc* and other genes in the septal network was far greater than expected by chance (Fisher's exact test, $p_{adj}$ = 6.97 × $10^{-5}$). Furthermore, analysis of network topology in Cytoscape showed that *Ubc* had the highest measure of *stress centrality* and *betweenness centrality*, key indicators of network hubs (*Shannon et al., 2003*).

Other genes with high centrality scores were *Akt1* and *Prkaca*, encoding AKT Serine/Threonine kinase/Protein Kinase B (AKT/PKB) and Protein Kinase A (PKA), respectively, which were downregulated in 129T2/SvEms compared to QSi5 septa, and had dispersed downregulated connections across the network. These connections were over-represented in their respective known and predicted phosphorylation targets (sourced from *PhosphoSitePlus Hornbeck et al., 2012* and *PhosphoPICK Patrick et al., 2015*) - 12/31 for ATK, 9/17 for PKA (P<1 × $10^{-4}$ for both kinases; Fisher's exact test) (*Figure 9*). We further analyzed known and predicted kinase targets among all genes in the network, which resulted in an additional 35 phosphorylation connections for PKA and nine for AKT. Interestingly, along with hubs for *Ubc*, collagen/ECM, and translation, an additional hub centered on *Src*, encoding a proto-oncogene tyrosine kinase involved in embryonic development and cell growth, was evident at E12.5, but not at later stages (*Figure 8—figure supplement 1*). Across all stages, there were very few DEGs in which predicted-pathogenic coding variants were identified, and none appeared as hub genes, placing network genes downstream of functional variants.

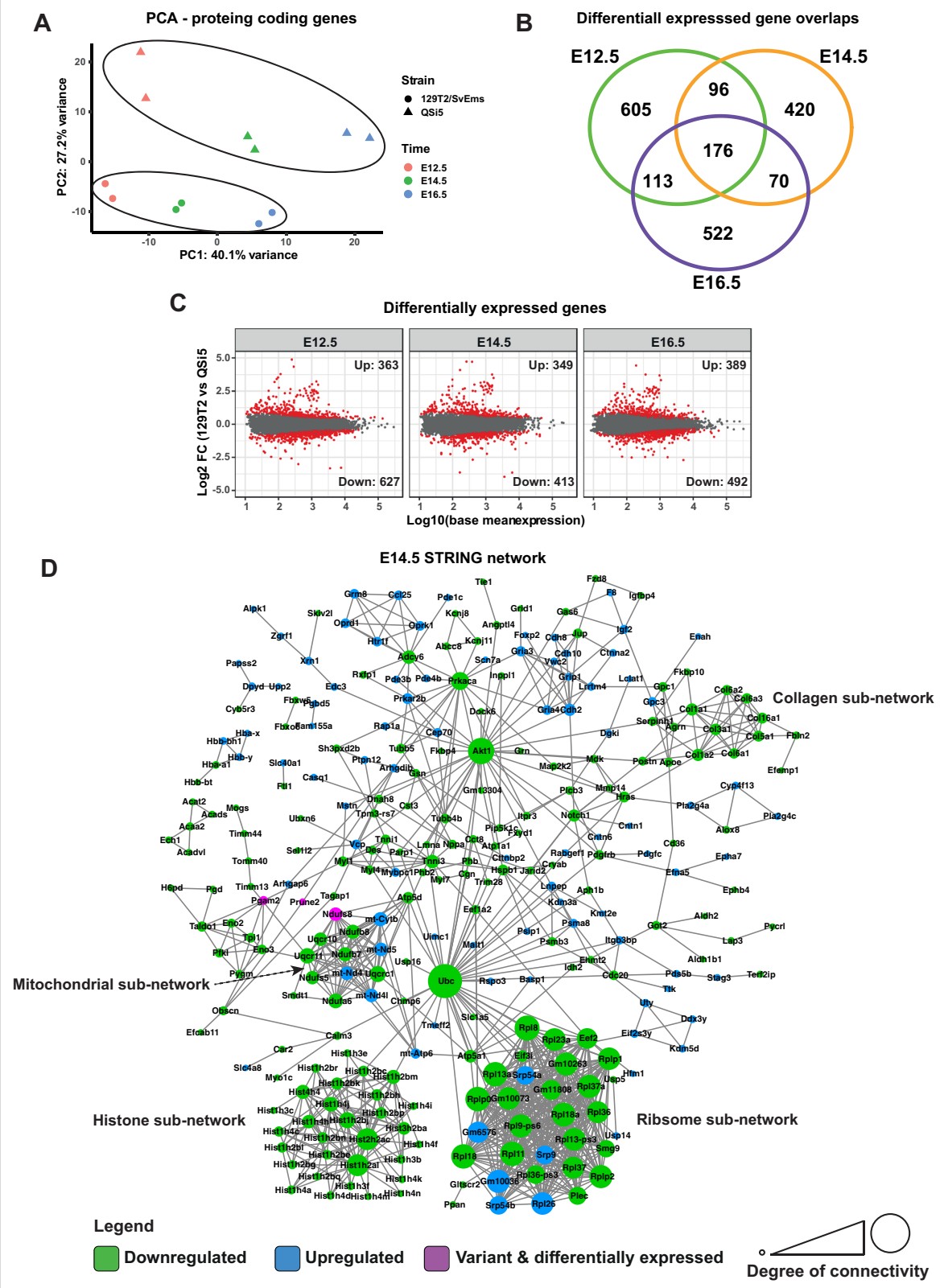

**Figure 8.** Transcriptome analysis of atrial septum. (**A**) Principal component analysis (PCA) plot of gene expression profiles from atrial septum RNA-seq libraries including two replicates per time point (E12.5, E14.5, and E16.5) per mouse strain (QSi5 and 129T2/SvEms) for protein-coding genes. (**B**) Venn diagram with the number of differentially expressed genes between QSi5 and 129T2/SvEms at different time points. (**C**) MA plots of differentially expressed genes for each time point. (**D**) *STRING* network of E14.5 differentially expressed genes and genes with predicted-pathogenic variants from

*Figure 8 continued on next page*

QSi5 or 129T2/SvEms. Genes are colored according to whether they are upregulated or downregulated in 129T2/SvEms or contain a variant that is predicted-pathogenic and differentially expressed.

The online version of this article includes the following figure supplement(s) for figure 8:

**Figure supplement 1.** STRING network of E12.5 differentially expressed genes and genes with predicted-pathogenic variants from QSi5 or 129T2/SvEms.

**Figure supplement 2.** STRING network of E16.5 differentially expressed genes and genes with predicted-pathogenic variants from QSi5 or 129T2/SvEms.

## Differentially expressed QTL genes are enriched for genomic variants

Both protein-coding and *cis*-regulatory variants are likely causally linked to network perturbations underlying quantitative traits, as highlighted by GWAS on human disease (*Zhang et al., 2014*). To explore the relationship between QTL variants and DEGs further, we assessed the enrichment of variants (129T2/SvEms vs QSi5) within and around DEGs underlying QTL relative to genomic features (enhancers, promoters, 5′UTRs, exons, introns, and 3′UTRs), comparing to variant distribution in non-DEGs. Enhancers were defined as regions showing enhancer histone (H) marks (H3K4me1$^+$ and/or H3K27ac$^+$; H3K4me3$^-$) within regions covering 50 Kb-250 Kb upstream and downstream of the transcriptional start site (TSS) for genes expressed in mouse embryonic stem cell-derived cardiac progenitor cells and cardiomyocytes, and whole E14.5 fetal hearts (*Wamstad et al., 2012*). We selected DEGs as protein-coding genes with an adjusted differential expression p-value ($p_{adj}$) of <0.05 and absolute log2 fold-change of >0.5 in at least one atrial septal time point. Of 2168 such genes, 214 were under QTL (1-LOD drop-off) (*Supplementary file 8*). For non-DEGs, we selected 2168 protein-coding genes

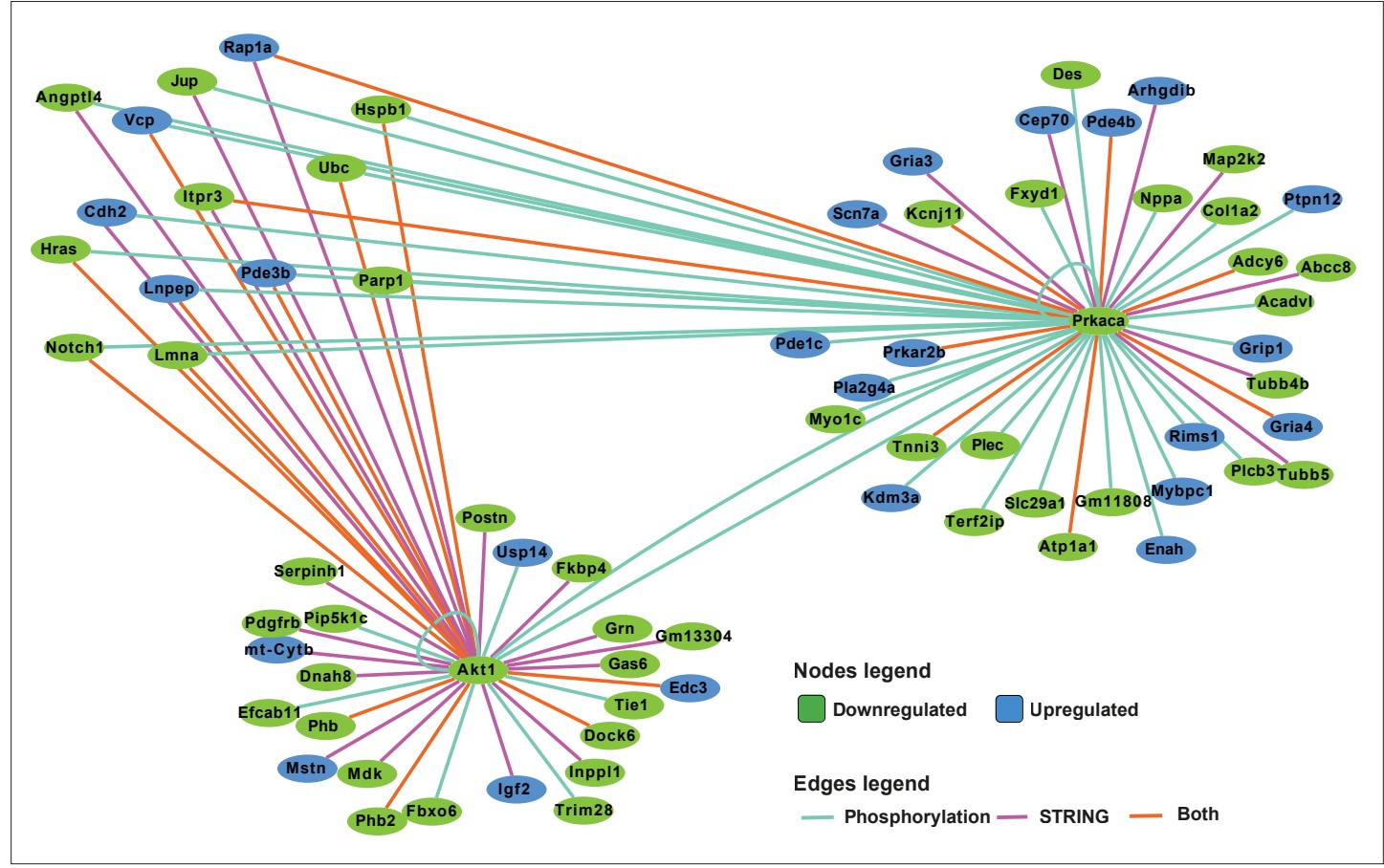

**Figure 9.** Network analysis of protein kinase A (PKA) and AKT1 phosphorylation targets. Integrated network analysis of PKA and AKT1 *STRING* connections and phosphorylation substrate targets from PhosphoSitePlus and PhosphoPICK predictions.

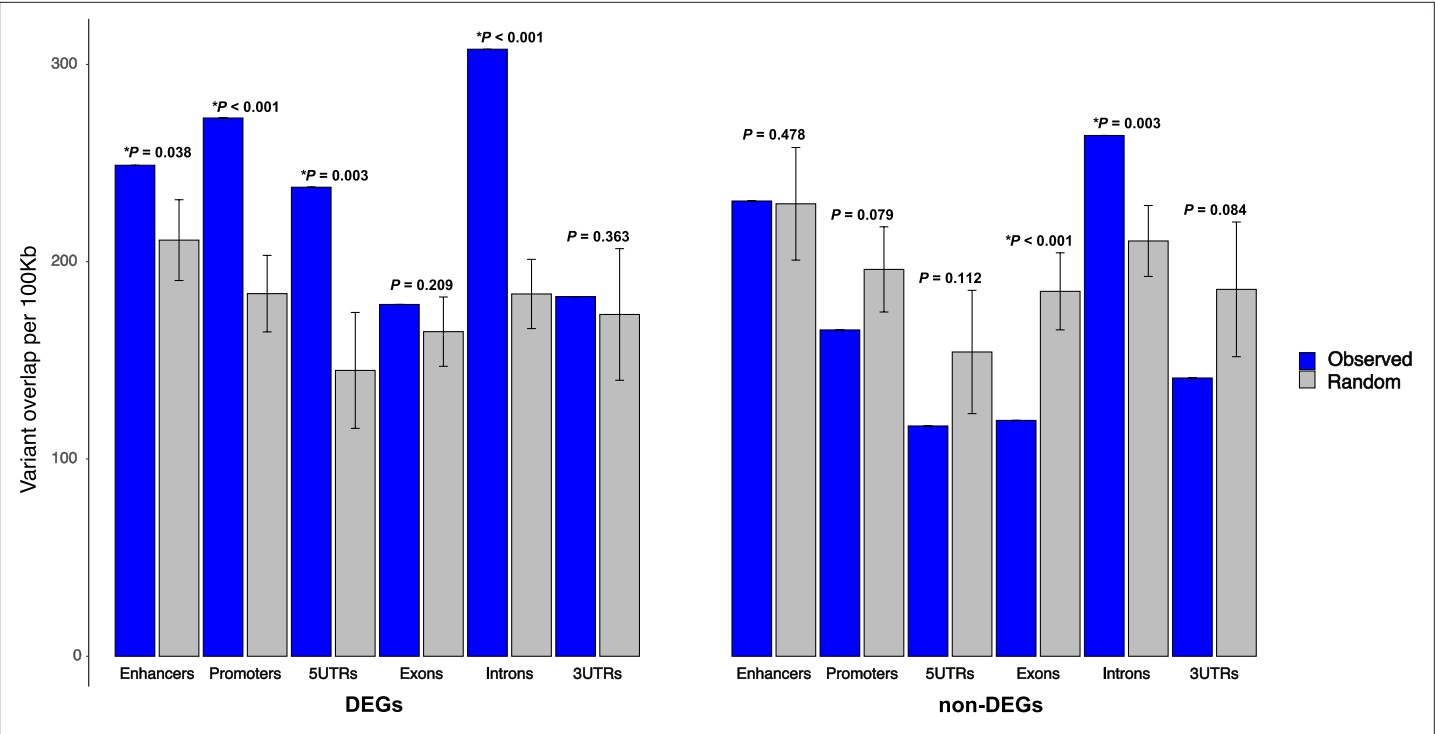

**Figure 10.** Enrichment of 129T2/SvEms vs QSi5 variants in genomic features of differentially expressed genes (DEGs) under quantitative trait loci (QTL). The number of variants directly overlapping a feature is shown in blue, and gray bars represent the expected values based on the mean overlap from 1000 randomly generated interval sets. Error bars show the 95% confidence intervals of the mean. The significance of the enrichment is expressed as p-values, calculated by dividing the number of random samples showing equal or greater overlap than the observed by the total number of permutations (*p<0.05).

The online version of this article includes the following figure supplement(s) for figure 10:

**Figure supplement 1.** Gene counts by number of 129T2/SvEms vs QSi variants.

**Figure supplement 2.** Impact of increasing enhancer window size on over-representation of variants in differentially expressed genes (DEGs) and non-DEGs under quantitative trait locus (QTL), and on number of enhancers detected.

that showed the highest $p_{adj}$ value after merging data from all three time points, of which 188 were under QTL (**Supplementary file 8**).

After permutation testing, we observed a significant enrichment of variants relative to expected in enhancers, promoters, 5′UTRs, and introns for DEGs underlying QTL (**Figure 10**). Non-DEGs showed significant depletion of variants in exons. DEGs and non-DEGs showed similar normalized variant count distributions across features (**Figure 10—figure supplement 1**), indicating that DEGs were not substantially skewed toward genes with high variant density. Likewise, variants were not focused on any specific gene class (**Supplementary file 8**). The over-representation of variants was consistent when candidate cardiac enhancers near DEGs were considered even up to 250 Kb from the TSS (**Figure 10—figure supplement 2A**), noting however that most enhancers were found within the 100 Kb window (**Figure 10—figure supplement 2B**). Interestingly, DEGs had approximately 60–80% more enhancers than non-DEGs, consistent with their higher levels of expression, irrespective of septal stage or strain (**Figure 10—figure supplement 2C–E**).

## Discussion

QTL analysis has emerged as an approach to understand the genetic complexity underpinning both quantitative and complex (non-Mendelian) binary traits. PFO and ASD are examples of complex binary traits of medical significance. One model for complex binary traits assumes an underlying continuous but unobservable variable (termed liability) with a threshold above which an individual expresses a phenotype (**Falconer, 1965**). Quantitative parameters act as proxies for the assumed

liabilities and significantly increase the power of QTL detection. Comparing F2 and AIL designs with identical parameters including sample size and marker density, the predicted confidence intervals of AIL QTL are $t/2$ times smaller than those of F2 QTL, where $t$ is the number of AIL generations (**Darvasi and Soller, 1995**). This indicates that AIL is a powerful method for precise localization of QTL and also separation of linked QTL identified by an F2 design.

We previously mapped QTL underlying quantitative parameters of the atrial septum using an F2 intercross design and here applied the AIL approach for confirmation and fine mapping. The cascade breeding program generating the F14 AIL resource from 48 breeding pairs per generation came close to the practical optimum (100 animals produced per generation) (**Darvasi and Soller, 1995**). From seven F2 QTLs included in our AIL study, at least six were confirmed and substantially narrowed, whereas five F2 QTLs resolved into multiple peaks, and additional QTLs were discovered. Overall, 37 significant atrial septal QTLs were documented.

Among the three quantitative parameters studied here, FVL and FOW showed a similar pattern of linkage on most of the chromosomes. This is a new finding that was not evident in the F2 study. Furthermore, independent analysis of PFO as a binary trait strongly supported results for FVL and/or FOW on most chromosomes. As noted, of the traits analyzed, FVL has a larger variation between parental strains (**Table 1**; **Kirk et al., 2006**) and shows a stronger negative correlation to PFO prevalence among several inbred and mutant strains (**Biben et al., 2000**), as well as in both F2 and F14 mice. Collectively, these data suggest that many QTLs affect the formation of the primary and secondary atrial septa in common, and that FVL is a robust indicator for atrial septal morphology and risk of PFO.

CRW did not show a significant correlation to PFO in the AIL data and was, therefore, not found to be a satisfactory surrogate for the size of the open corridor in cases of PFO (**Biben et al., 2000**). Nonetheless, we found five significant QTLs for CRW, only one of which overlapped with QTL for the other traits, suggesting that CRW is determined largely by different genetic elements to those governing FVL and FOW. As CRW is measured along the lower boundary of the apoptotic domain that generates the *ostium secundum* in septal development, it may relate to tissue remodeling subsequent to the apoptotic process.

As septal parameters may be influenced by heart size, we performed a linkage analysis for normalized HW across MMU11, where a significant QTL was discovered retrospectively in the F2 data, and indeed across all other chromosomal regions for which markers were selected. We confirmed and refined the position of the HW QTL on MMU11, and detected HW QTL on MMU2, MMU4, MMU9, and MMU13. Of these, only MMU2 has previously been linked with HW (**Rocha et al., 2004**). Inter-trait correlation and linkage results showed that HW influences quantitative septal parameters in both F2 and F14 studies, but only in a relatively minor way (**Supplementary file 2**). Importantly, HW and atrial septal morphology QTL showed limited overlap. We propose that HW is mostly determined by ventricular chamber growth as an independent parameter to the development of 'primary' myocardial components of the early heart tube, which have a lower proliferative index (**Moorman and Christoffels, 2003**), and that of the mesenchymal and cushion components, that contribute to the formation of the inter-atrial septum.

Overall, we conclude that septal morphological variation and risk of PFO have a complex genetic basis in inbred strains of mice. Given that the analysis was restricted to the previously found QTL and the selected markers covered only a limited part of the genome, it is likely that the genetic complexity underpinning septal defects in the two inbred strains under study is even greater than revealed here. It is noteworthy that the two inbred lines in this study were previously selected for independent traits. For example, the QSi5 strain, used here because of its low PFO risk (**Kirk et al., 2006**), was bred for numerous traits related to high fecundity (**Holt et al., 2004**) and whereas this has facilitated the generation of >1000 mice required for the AIL study, the genetic diversity that contributes to variation in atrial septal morphology may be limited by the prior genetic selection. High genetic complexity underpinning variation in atrial septal morphology may be expected, given the diverse lineage origins, tissue contributions, and morphogenetic networks contributing to septal structure (**Anderson et al., 2003**; **De Bono et al., 2018**; **Deepe et al., 2020**; **Rana et al., 2014**; **Steimle et al., 2018**). Extrapolating to the diverse outbred human population, we might conclude that potentially many hundreds of variants contribute to atrial septal dysmorphology, consistent with estimates of the number of inherited and de novo mutations discovered to date contributing to CHD more broadly (**Jin et al., 2017**).

To create a list of protein-coding genes that could potentially contribute to differences in atrial septal morphology in inbred strains of mice, we sequenced the whole genomes of parent strains and then filtered for genes that lie under QTL (1-LOD drop-off), were expressed in the developing inter-atrial septal septum, and carried variants predicted to be pathogenic. The total list comprised 61 genes carrying 92 variants, resulting in an average of ~2.5 candidate protein-coding variants per QTL. Most of these genes lay within regions of high SNP density and 26% of them carried multiple variants. We further filtered for genes that overlapped the MGI list of mouse genes associated with cardio-vascular phenotypes, and a curated known and emerging human CHD gene list (*Alankarage et al., 2019*). Eight genes (*Cybrd1*, *Dst*, *Fxn*, *Lrp2*, *Mcam*, *Pgbd1*, *Sik3*, *Smad6*) overlapped with curated lists. The detailed role of most of these genes in septal development has not been elucidated. Our follow-up study of the SMAD6 linker region variant R282P demonstrated a hyper-ubiquitinated state for this critical regulator of BMP signaling. The full mechanism and in vivo significance remain to be determined, noting that hyper-ubiquitination, in addition to affecting SMAD6 stability in vivo, could inhibit additional regulatory processes or interactions occurring across the linker region (*Sapkota et al., 2007*; *Xu et al., 2016*).

Extending genome analyses, we determined the transcriptome of atrial septal regions of both strains at three developmental time points and calculated DEGs. By far the most significant *STRING* protein network for DEGs was at E14.5, with subnetworks with high connectivity associated with growth-related macromolecular cellular structures including nucleosomes, ribosomes, mitochondria, and ECM. Virtually all associated genes were downregulated in the 129T2/SvEms strain (showing less robust septal characteristics). Our findings suggest that there is a general septal growth and matura-tion deficit involving multiple septal components in the 129T2/SvEms strain at a relatively late stage (E14.5) of septal development. This idea is supported by the correlations between FVL and FOW (reflecting the development of the *primum* and *secundum* septa, respectively) (*Kirk et al., 2006*) and the strong concordance between QTL for individual septal traits and PFO discovered in this work. We hypothesize that differences in quantitative septal traits and PFO prevalence that we documented previously in an inbred, hybrid, and mutant strains of mice (*Kirk et al., 2006*), relate in part to degrees of septal growth and maturation, and that higher septal growth and maturation facilitates better fusion of the *septum primum* and *secundum*. Careful elucidation of the differential growth parameters of septal components in parent strains across time may provide further support for this model.

*STRING* networks identified three potential hub drivers of this depressed biosynthetic state in the 129T2/SvEms strain - genes for Ubiquitin c, and kinases AKT and PKA. Many genes encoding targets of these kinases were also up- or downregulated, potentially a network adaptation. We propose that variants in the 129T2/SvEms parent strain depress growth-related signaling pathways (including BMP, AKT, PKA, and SRC) and associated ubiquitin-related quality control processes, impacting septal growth and morphogenesis independently of chamber growth. In this light, it is interesting that CHD and cancer risk genes have recently been correlated (*Morton et al., 2021*). This could be a focus area for further investigation.

Our analysis of variant architecture under QTL (*Figure 10*) showed that variants (129T2/SvEms vs QSi5) were over-represented in differentially expressed atrial septum-expressed genes located under QTL across multiple gene features including enhancers, promoters, 5'-UTRs and introns. These data suggest a direct mechanistic link between the presence of non-coding (including cis-regulatory) as well as coding region variants in the differential expression of septal genes underlying QTL. This would represent a significant departure from a reductionist model whereby a single variant accounts for the impact of each QTL. Cis-regulatory variants are, therefore, candidate drivers of network pertur-bations revealed by *STRING* analysis.

Our results are consistent with Fisher's 'infinitesimal' model whereby many loci (theoretically an infinite number) could each contribute to a small part of the liability for common disease (*Fisher, 1919*; *Norton and Pearson, 1976*), and also GWAS analysis of human disease that most often detect common risk loci of small effect. However, as in our previous study (*Kirk et al., 2006*), individual septal QTL can have significant effect sizes, demonstrating the power of the AIL approach to discern alleles of major impact. Indeed, all five predicted pathogenic coding variants highlighted in this study, including the *Smad6* variant, lay under QTL of moderate to high effect size (5.68–65%). It is possible that a limited number of high-impact variants are responsible for septal defects in certain AIL mice. This type of polygenic inheritance has been demonstrated recently in a human cardiomyopathy family

(*Gifford et al., 2019*). Such variants may prove to have causative or modifying effects on atrial septal defects in humans, as exemplified by *SMAD6*.

Several findings speak to the complex genetic landscape of PFO. It is interesting that twelve out of the 37 QTL detected were cryptic QTL, with many being of moderate to large effect size (~5–70%; *Supplementary file 3*). Thus, cryptic QTL is frequently detected and individually appear to provide strong buffering effects against septal defects. Cryptic QTL has been detected commonly in other animal and plant QTL studies (*Rieseberg et al., 1999*). The mechanisms of their action would be interesting to identify as they may reveal pathways that protect against CHD more broadly.

A further complexity is that, of five genes carrying predicted pathogenic variants highlighted in this study, four were found in the unexpected strain. For example, *Cep164* lies under a cryptic QTL (protective), whereas predicted pathogenic variants in *Cep164* were found in the 129T2/SvEms strain (showing worse septal status). The inverse can also apply – for example, *Lrp2* lies under a normal QTL, but variants were found in the QSi5 strain. Such findings suggest highly complex interactions within and between individual QTLs. As noted, 26% of predicted pathogenic genes, some known to be involved in heart development, carried multiple predicted pathogenic variants (*Cep164* carried seven). It would be reasonable to imagine that these alleles are null or have major impacts on gene function. Genetic compensation may also buffer the impact of such variants (*El-Brolosy et al., 2019*).

Our results provide the first high-resolution picture of genetic complexity and multilayered network liabilities underpinning atrial septal variation in a mouse model. Elucidating the impacts of genetic variants under QTL at the individual gene level would be a daunting task; however, understanding higher level network control of septal development underpinning observed defects in septal macro-molecular synthesis pathways is an important cardiac developmental systems genetics problem, which may have relevance to other congenital dysmorphologies. Our quantitative model offers opportunities to study the interface between genetic, environmental, and epigenetic inputs to common CHD and potentially other diseases in greater detail.

# Materials and methods

**Key resources table**

| Reagent type (species) or resource | Designation | Source or reference | Identifiers | Additional information |
|---|---|---|---|---|
| Strain, strain background (*M. musculus*) | 129 T2/SvEms | The Jackson Laboratory *Biben et al., 2000* | Stock no. 002064 | http://www.informatics.jax.org/mgihome/nomen/strain_129.shtml |
| Strain, strain background (*M. musculus*) | QSi5 | The Jackson Laboratory | Stock no. 027001 | |
| Antibody | Smad6 (mouse monoclona) | Santa Cruz Biotechnology | SC-25321; RRID: AB_627906 | WB (1:100) |
| Antibody | FLAG-M2 (rabbit monoclona) | Cell Signaling Technology | CST:14793; RRID: AB_2572291 | IP (1:150) WB (1:1000) |
| Antibody | Ubiquitin (mouse monoclona) | Santa Cruz Biotechnology | SC-166553; RRID: AB_2241297 | WB (1:1000) |
| Cell line (*H. sapiens*) | HEK293T | ATCC | RRID: CVCL_0063 | |
| Transfected construct (*H. sapiens*) | PCS2-SMAD6 | ADDGENE | RRID:Addgene_14960 | |
| Transfected construct (*M. musculus*) | BRE-luc | ADDGENE | RRID:Addgene_45126 | |
| Peptide, recombinant protein | BMP4 | Gibco | Gibco:PHC9534; | 100 ng/mL |
| Commercial assay or kit | Dual luciferase assay kit | Promega | Promega:E1980 | |
| Commercial assay or kit | KAPA hifi | Roche | Roche:KK2101 | |
| Commercial assay or kit | WesternBreeze Chemiluminescent kit-Anti-rabbit | Thermo Fisher Scientific | Thermo Fisher:WB7106 | |
| Software/ algorithm | ImageJ | NIH | RRID:SCR_003070 | |

*Continued on next page*

*Continued*

| Reagent type (species) or resource | Designation | Source or reference | Identifiers | Additional information |
|---|---|---|---|---|
| Software/ algorithm | Graphpad Prism | Dotmatics | RRID:SCR_002798 | |
| Other | Recombinant protein G-Sepharose 4B | Thermo Fisher Scientific | Thermo Fisher:101241 | |
| Software/algorithm | R | https://cran.r-project.org | | Version 4.2.2 |
| Software/algorithm | GenomicRanges (R package) | *Lawrence et al., 2013* | | Version 1.42.0 |
| Software/algorithm | GenomicFeatures (R package) | *Lawrence et al., 2013* | | Version 1.42.3 |
| Software/algorithm | liftOver | Kent tools via UCSC | | version 402 |
| Software | Trimmomatic | *Bolger et al., 2014* | | v0.35 |
| Software/Algorithm | STAR aligner | *Dobin et al., 2013* | | v2.5.1 |
| Software/Algorithm | Salmon | *Patro et al., 2017* | | |
| Software/Algorithm | DESeq2 | *Love et al., 2014* | | v1.14.1 |
| Software/algorithm | PANTHER | *Mi et al., 2017* | | Web service |
| Software/algorithm | BWA-MEM | *Li and Durbin, 2010* | | v0.7.15 |
| Software/algorithm | GATK | van der Auwera and O'Connor, 2020 | | v3.5 |
| Software/algorithm | VCFtools | *Danecek et al., 2011* | | v0.1.14 |
| Software/algorithm | Annovar | *Wang et al., 2010* | | |
| Software/algorithm | Cytoscape | *Shannon et al., 2003* | RRID: SCR_003032 | V3.9.1 |
| Software/algorithm | Python | https://www.python.org | | |
| Software/algorithm | Networkx | *Hagberg et al., 2008* | | V3.0 |
| Other | STRING | *Szklarczyk et al., 2017* | | v10 |
| Software/algorithm | Spliceogen | *Monger et al., 2019* | | |
| Software/algorithm | Variant Effect Predictor | *McLaren et al., 2016* | | |
| Software/algorithm | SIFT | *Ng and Henikoff, 2003* | | |
| Software/algorithm | fgsea | https://doi.org/10.1101/060012 | | v1.22.0 |
| Other | Mouse Molecular Signatures Database | https://www.gsea-msigdb.org/ | | version m5.all.v2023.1.Mm |

## QTL fine mapping using advanced intercross line

### Mice and advanced intercross line

129T2/SvEms mice (sub-strain 129T2/SvEms-+[Ter]?) were obtained from The Jackson Laboratory and were a derivative of the Ter subline family of 129 strains (*Biben et al., 2000*). QSi5 is an inbred mouse strain that was bred from outbred Quackenbush Swiss mice (*Holt et al., 2004*) at the University of Sydney, Sydney NSW, and was maintained as a breeding colony once established. QSi5 mice are also available from the Jackson Laboratory (strain 027001). As parental strains in our F2 study (*Kirk et al., 2006*), 129T2/SvEms and QSi5, they were selected based on having extreme values for mean FVL and prevalence of PFO, two indicators of atrial septal morphology which are strongly negatively correlated (*Biben et al., 2000*). 129T2/SvEms had the highest prevalence of PFO (75%) and shortest FVL (mean = 0.6 mm) among inbred strains and multiple crosses (*Biben et al., 2000*). QSi5 had the longest FVL (mean = 1.13 mm) and among the lowest prevalence of PFO (4.5%).

We randomly inter-crossed the original F2 mice for 12 further generations. The F2 mice were bred to produce 48 male and female pairs which were then stocked in 48 separate cages. We inter-crossed

the mice in each cage to generate F3 mice. For the F3 × F3 and subsequent crosses, a cascading scheme was used in which a female mouse from one cage would be mated with a male mouse from the next cage. This breeding design, in which each pair contributes exactly two offspring (one male and one female) to the next generation doubles the effective population size and reduces the random changes in allele frequency due to random genetic drift over the 10 generations. Animals were bred and housed under Animal Care and Research Ethics approvals N00/4-2003/1/3745, N00/4-2003/2/3745 and N00/4-2003/3/3745 from the University of Sydney.

## Dissection and measurements

In total, 1003 AIL F14 mice were dissected. Of these, 933 had complete phenotypic data (475 males and 458 females). Phenotyping of each mouse including initial and fine dissections, determination of PFO status, and measurement of septal features, were all performed on the same day. As in the F2 study (*Kirk et al., 2006*), the thoracic organs including the heart, lungs, and mediastinum were initially dissected *en bloc* and stored in PBS. A tail biopsy was also taken from each mouse and snap-frozen for DNA extraction.

The following steps were performed under a Leica MZ8 dissecting microscope. The mediastinal organs were removed to expose the atria. Subsequently, the left atrium was opened to expose the atrial septum. We detected PFO by pressurization of the right atrium. The right-to-left passage of blood (or injected Orange G dye) across the inter-atrial septum indicated the presence of PFO. Measurement of septal features including FVL, FOW, and CRW was performed using an eyepiece graticule. FVL was defined formally, as in our previous study (*Kirk et al., 2006*), as the length of the flap valve from the edge of the crescent (proximal rim of the *ostium secundum*) to the distal rim of the fossa ovalis. The maximum width of the *foramen ovale* (*foramen ovale* width; FOW) was measured perpendicular to the FVL. Crescent width (CRW) was defined as the maximum width of the prominent crescent-shaped ridge, representing the proximal rim of the *ostium secundum* and edge of the flap valve as previously described (*Kirk et al., 2006*).

## Normalization

A general linear model (PASW Statistics 18) was used to analyze the effect of various covariates on the traits of interest (FVL, FOW, and heart weight; HW) considering a p-value of <0.05 as significant (*Supplementary files 9-13*). FVL and FOW were significantly affected only by HW (p=0.002 and p=0.028, respectively) and the effect of other covariates including sex, age, body weight (BW), and coat color was not significant. However, we did not adjust for HW since QTL relevant to HW may also influence atrial septal morphology. On the other hand, HW was significantly affected by sex (p<0.001), age (p=0.032), BW (p<0.001), and coat color (p=0.042). Therefore, prior to sample selection and further analysis, HW was normalized for age, sex, and BW, although not for coat color so as to avoid missing QTL linked to coat color genes.

## Sample selection

Selective genotyping of extreme phenotypes is an efficient method to increase the power of QTL mapping (*Lander and Botstein, 1989*). However, the benefits of this method decline with an increasing number of uncorrelated or weakly correlated traits. The focus of our study was to fine-map QTL underlying the main highly correlated traits (FVL and FOW) and a peripheral trait of HW. We did not consider CRW as a basis for sample selection since it was a less defined anatomical structure and, unlike FVL and FOW, it was not associated with PFO (see Results). For each trait (FVL, FOW, and HW), we selected approximately 100 F14 animals with extreme phenotypes. Given the overlap between the extreme phenotypes from different traits, 237 mice were selected by this method. To compensate for biases in selective genotyping in this study, the selection of extreme phenotypes was combined with a degree of random selection. Therefore, 163 mice were also selected randomly giving a total sample of 400 mice. The same number of males and females were selected from the breeding cages. Thus, the selected mice gave as equal as possible representation of males, females, and breeding cages.

## Marker selection

We searched genotype data from the Mouse HapMap project for potential informative single nucleotide polymorphisms (SNPs) between 129T2/SvEms and QSi5 strains. In total, a set of 135 markers with an average interval of 2 centimorgans (cM) was selected to genotype genomic regions including the significant QTL from the F2 study (three QTL for FVL and three for FOW) and one QTL for HW (*Supplementary file 14*). In addition, a suggestive QTL for FOW on MMU9 (LOD = 3.43) was included in the AIL study as its peak covered the known ASD/VSD gene *Tbx20* (*Kirk et al., 2006*). Fifteen extra markers were chosen to cover the whole peak of this QTL. Subsequently, a total of 150 markers were genotyped in the selected mice (*Supplementary file 14*).

## Genotyping

Genomic DNA was isolated from mouse tails using a Macherey-Nagel Nucleospin kit according to the manufacturer's guidelines. Prior to genotyping, we verified the informativeness of the markers for parental strains. Subsequently, the markers were genotyped in the selected mice using iPLEX MassArray assay following the manufacturer's protocol.

## Linkage analysis

We performed interval mapping linkage analysis for the quantitative traits at 0.25 cM intervals using a maximum likelihood method implemented in R software (see code availability statement below). The method used was initially developed for an F2 design, but modified for an AIL using the methods described by *Darvasi and Soller, 1995*. Given the density of selected markers and the size of the genomic region covered by markers, a LOD score of 2 was set as the threshold level of significance (*Lander and Botstein, 1989*).

We used the 1-LOD drop-off to estimate the confidence interval of each QTL (*Lander and Botstein, 1989*). However, in some cases, two significant peaks were located close together and overlapped in the 1-LOD drop-off intervals. To determine whether these peaks reflected the same underlying QTL or identified the presence of independent QTL, we re-ran the linkage analysis using a model in which the marker closest to the higher peak was included as a fixed term (*Figure 3—figure supplement 1*), effectively a simplified form of composite interval mapping but applied in a maximum likelihood framework (*Kearsey and Hyne, 1994*; *Wu and Li, 1994*). Disappearance of the lower peak would indicate that two peaks represented a single QTL. On the other hand, if the lower peak remained significant, two peaks would represent two separate QTLs.

We have previously developed a QTL program in R to perform linkage analysis for PFO as a binary trait (presence or absence) (*Moradi Marjaneh et al., 2012*). For binary analysis of AIL data, the linear model used for the quantitative analysis was replaced by a generalized linear model in the form of a logistic regression model, as described previously (*Moradi Marjaneh et al., 2012*).

Additive and dominance effects given in *Supplementary file 3* were used to calculate QTL effect size information. The additive effect for each QTL (a.qtl) was defined as the effect of one copy of the Q (QSi5) allele and calculated as (mean.QQ - mean.qq)/2 where mean.QQ is the phenotypic mean for genotype QQ and mean.qq is the phenotypic mean for genotype qq (129T2/SvEms allele). The dominance effect (d.qtl) for each QTL was defined as the difference between the mean.Qq (phenotypic mean for genotype Qq) and the midpoint between the qq and QQ means. Therefore, if d.qtl >0, allele Q is (partially) dominant as Qq is moved towards QQ and if d.qtl <0, allele q is (partially) dominant as Qq is moved towards qq. Attributable phenotypic variance for each QTL is the additive effect of the QTL, expressed as a percentage of the difference between parental means for that trait.

## Conversion of genetic maps

We previously defined the genetic positions of the F2 markers using an older mouse genetic map developed by the Whitehead Institute and the Massachusetts Institute of Technology (MIT) (*Dietrich et al., 1996*). Thus, we converted the genetic position of F2 markers into those for the current Mouse Genome Informatics (MGI) map developed at the Jackson Laboratory (*Bult et al., 2008*) using mouse maps introduced by *Cox et al., 2009*. For inter-marker intervals, a linear interpolation was used to convert old genetic positions to new ones.

## Whole genome sequencing of the AIL parental strains

### Sequencing

Genomic DNA was extracted from tail biopsy specimens of QSi5 and 129T2/SvEms mice using the phenol-chloroform extraction method. Library preparation and sequencing were performed at the Kinghorn Centre for Clinical Genomics. 200 ng of DNA was mechanically fragmented using LE220 Covaris (Covaris, Woburn, USA) to approximately 450 bp inserts followed by library preparation using the Seqlab TruSeq Nano DNA HT kit (20000903, Illumina, San Diego, USA). Library preparation was performed according to the manufacturer's instructions which included end repair, library size selection, 3' end polyadenylation, adaptor ligation, purification of ligated fragments, enrichment of ligated fragments by PCR, and a final purification of amplified DNA library. Library fragment sizes were reviewed using the LabChip GX (Perkin Elmer, Waltham, USA) to evaluate library size and adapter dimer presence before the library concentration was determined via quantitative real-time PCR using the KAPA library quantification kits (KK4824, Roche, Basel, Switzerland) on the QuantStudio7 or ViiA7 (Thermo Fisher Scientific, Waltham, USA). Normalized DNA libraries were clustered on Illumina cBot and then sequenced using Illumina HiSeq X Ten platform using HiSeq X Ten Reagent Kit v2.5 kits (FC-501–2501, Illumina). Paired-end sequencing was performed using the 2 × 150 bp chemistry to achieve an average output of approximately >120 Gb of data per library.

### Bioinformatic analysis

The sequencing reads were mapped to the mouse reference genome (NCBI38/mm10) using Burrows-Wheeler Aligner (BWA-mem v0.7.15) (*Li and Durbin, 2010*). Variants were called using the genome analysis toolkit (*GATK*; v3.5) pipeline following best practice guidelines (*van der Auwera and O'Connor, 2020*). This involved marking duplicate reads with the *Picard* toolkit (Broad Institute GitHub Repository: http://broadinstitute.github.io/picard/) (RRID:SCR_006525 version 2.21.3), indel realignment with GATK using known SNPs and indels from C57BL/6 dbSNP142 (ftp://ftp-mouse.sanger.ac.uk/current_indels/strain_specific_vcfs/), Base quality score recalibration and final variant calling was performed using the GATK Haplotype Caller. Variant call format (VCF) files were merged using *VCFtools* (v0.1.14) (*Danecek et al., 2011*) and annotated with ANNOVAR (*Wang et al., 2010*). A set of high-quality variants was identified by filtering variants for those with a minimum sequencing depth of eight and a maximum estimated false discovery rate (FDR) of 10%. Variants were filtered to only consider those that were homozygous in one mouse line with either the reference or a heterozygous call in the other line. Zygosity information from each mouse line was extracted using *VCFtools* and custom scripts.

To analyze protein-coding variants, variants were filtered for feature annotations 'exonic,' 'exonic;splicing,' or 'splicing,' with variants annotated with effect class 'synonymous SNV' filtered out. Variants were filtered for genes expressed in the septal time course RNA-seq (see below), where a gene was defined as expressed if it had a counts-per million (CPM) >1 in at least two samples. They were then converted to mm10/GRCm39 coordinates and submitted to the Ensembl Variant Effect Predictor (VEP) tool (*McLaren et al., 2016*). Based on VEP annotations, variants were classified as high-impact (VEP impact classification of 'HIGH'), deleterious missense (*SIFT* (*Ng and Henikoff, 2003*) 'deleterious' prediction), or non-high-impact deletions. High-impact variants were manually inspected in IGV and filtered out if the 'HIGH' impact classification was deemed inaccurate (for example, cases where an adjacent variant negated a predicted stop gain or a substitution was incorrectly called as a deletion). Finally, the list of high-confidence potential-pathogenic variants was filtered for variants falling within QTL coordinates (LOD >1).

## Transcriptome analysis of cardiac interatrial septum in mice

Microdissection of the atrial septum region was performed on embryos from mouse strains (QSi5, 129T2/SvEms) at E12.5, E14.5, and E16.5 (six septa per mouse strain per time point). Embryos were dissected from the uterus in ice-cold PBS under a dissecting scope (Leica MZ8) using micro-tweezers. Once the heart was harvested from the embryo and the pericardial membrane removed, the atrial and ventricular regions were separated. Then, the atrial appendages were removed from the atrial part and the remaining mesenchymal tissue from the atrioventricular canal was further trimmed. The remaining tissue, containing the atrial septum, was stored at –80 °C in RNAlater solution (Ambion).

Total RNA was purified using miRNeasy Micro Kit (Qiagen). Two technical replicates per mouse strain per time point were created by pooling RNA from three dissected septa each.

## RNA sequencing and data analysis

RNA libraries were made using TruSeq stranded RNA Library Preparation Kit (Illumina) and sequenced on a HiSeq2500 Illumina sequencer to a depth of ~23 million paired-end reads per sample. Sequencing reads were trimmed to remove poor quality sequence and adaptors using *Trimmomatic* (v0.35) (*Bolger et al., 2014*) using parameters LEADING:3 TRAILING:3 SLIDINGWINDOW:4:15 MINLEN:33. Sequencing reads were aligned against the mouse reference genome (GRCm38/mm10) using RNA STAR (v2.5.1) (*Dobin et al., 2013*). Gene counts were assembled using *SummariseOverlaps* (v1.30.0) (*Lawrence et al., 2013*) against Gencode release M4 and filtering to only include those whose expression was at least 1 count per million (CPM) in at least two samples. Differential expression analysis of RNA-seq data was performed using *DESeq2* (v. 1.14.1) (*Love et al., 2014*). For differential expression analysis, each time point was analyzed individually. Differentially expressed genes were calculated between 129T2/SvEms and QSi5 using the *DESeq* function with default parameters, with an adjusted p-value cut-off of 0.05 and absolute log2 fold-change difference of 0.5 used as thresholds for significance. GO term analyses of differentially expressed genes were performed using the *PANTHER* web service (*Mi et al., 2017*) with a false discovery rate cut-off of 0.05 used to assess significance. Transcript isoform quantifications were performed on the Fastq files using Salmon (*Patro et al., 2017*), against the Gencode mm10 transcript reference (vM10) with default parameters. Transcript abundances were read into DESeq2 using the tximport package (*Soneson et al., 2015*). Transcripts were filtered and retained for differential expression analysis if they had a CPM >1 in at least four samples across the strains and time points. DESeq2 testing of differences between strains while accounting for time point was performed using a design of '~strain + time point'.

## Network analysis

Network analysis of DEGs was performed using protein-protein association connections obtained from the *STRING* (*Szklarczyk et al., 2017*) version 10 database, considering connections with a combined score greater than 700. Permutation testing (100,000 permutations) was applied to determine whether the size and complexity of the networks (determined by the number of edges and clustering coefficient, respectively) were greater than expected by chance. For each permutation, a random set of genes was selected from the set of expressed genes in the RNA-seq for the relevant time point, and network metrics re-calculated. Edge counts and clustering coefficients were calculated using *NetworkX* (*Hagberg et al., 2008*). Empirical p-values were then determined as the proportion of random networks achieving network metrics equal to, or higher, than the DEG networks.

## Splicing analysis of *Tbx20*

*Tbx20* variants were examined for splicing defects using Spliceogen (*Monger et al., 2019*) with default parameters. To evaluate potential splice differences between the strains from the RNA-seq we used the splice junction (SJ) files produced by the STAR 2-pass alignment (*Dobin et al., 2013*). For each time point, we identified junctions overlapping the *Tbx20* gene (matched for strand) that were supported by over 10 uniquely mapped reads in both strains. The junction coordinates were then compared between the strains using the GenomicRanges *findOverlaps* function (*Lawrence et al., 2013*), with the parameter *type* set to 'equal'.

## Analysis of SMAD6 variant

### Plasmids

Pcs2-SMAD6 (#14960) and BRE-luc (#45126) were purchased from Addgene. FLAG tag was added to SMAD6 using the following primers:

> Forward (F): 5′GATCGACTACAAGGACGACGATGACAAGG 3′;
> Reverse (R): 5′GATCCCTTGTCATCGTCGTCCTTGTAGTC 3′.
> ***Mutagenesis primers***: R281P - F 5′CTCCCTACTCTCCGCTGTCTCCTCG 3′;
> R 5′CGAGGAGACAGCGGAGAGTAGGGAG 3′;
> Y279A - F 5′CTCCGCCACCTCCCGCATCTCGGCTGTCTC 3′;

R 5'GAGACAGCCGAGATGCGGGAGGTGGCGGAG 3';
G471S - F 5'CATCAGCTTCGCCAAGAGCTGGGGGCCCTG 3';
R 5'CAGGGCCCCCAGCTCTTGGCGAAGCTGATG 3'.

## Cell lines
HEK293T cells (ATCC; https://www.atcc.org; short non-tandem repeat authentication January 19, 2023, 100% match to reference; mycoplasma-negative) were used.

## Cell culture
HEK293T cells were maintained in a DMEM medium containing 10% FCS in a humidified incubator at 37 °C at 10% $CO_2$. 80,000 cells were seeded in 12-well plates for luciferase assays and 100,000 cells were seeded in six-well plates for protein extraction. BMP4 ligand (Gibco, #PHC9534) was added to cells at 100 ng/ml. Transfections were carried out using Lipofectamine 3000 (Thermo Fisher Scientific) according to the manufacturer's instructions.

## Luciferase assay
HEK293T cells were transfected with 100 ng of luciferase reporter constructs, 150 ng of expression vectors or empty vector and 2.5 ng of the TK-Renilla. Cells were treated with BMP4 overnight on the day of transfection. Assays were performed 24 hr after transfection. Dual-luciferase assays were performed as per the manufacturer's instructions (Promega, #E1980). Firefly luciferase activity was normalized to Renilla luciferase activity.

## Co-IP and western blot
HEK293T cells were transfected with 1 µg of hSMAD6-WT, hSMAD6-R281P, hSMAD6-Y279A, or hSMAD6-G471S. Cells were treated with either DMSO or MG132 (Sigma Aldrich, #M7449) 8 hr prior to protein extraction. Cells were lysed 48 hr after transfection using whole cell extract buffer (20 mM HEPES, 420 mM NaCl, 0.5% NP-40, 25% glycerol, 0.2 mM EDTA, 1.5 mM MgCl2, 1 mM PMSF, and protease inhibitors). Cells were washed in PBS and lysed with 250 µl WCE buffer for 10 min on ice. Lysed cells were scraped off the six-well plates and homogenized with a 25 Gauge needle 10 times. The lysates were centrifuged for 30 min at 4 °C. The supernatant was precleared with Protein G (Thermo Fisher Scientific) for 1 hr at 4 °C, then incubated with anti-FLAG M2 antibody (Sigma Aldrich, #F1804) at 1:150 dilution or an equal amount of mouse IgG overnight at 4 °C. The lysates were incubated with Protein G beads for 2 hr at 4 °C, then washed in WCE buffer four times. Protein was eluted in 4 x sample buffer (Biorad) for 5 min at 95 °C and loaded onto TGX stain-free precast gels (Biorad). Western blots were carried out using the following antibodies: anti-ubiquitin (1:1000, Santa Cruz Biotechnology, sc-166553), anti-FLAG (1:1000, Cell Signaling Technology, #14793), anti-SMAD6 (1:100, Santa Cruz Biotechnology, sc-25321).

## Variant enrichment analyses
We examined the overlap of high-quality homozygous 129T2/SvEms vs QSi5 variants and genomic features (enhancers, promoters, 5'UTRs, exons, introns, and 3'UTRs) of DEGs and non-DEGs located under QTL 1-LOD drop-off regions. Promoters were defined as 1000 bp upstream and 100 bp downstream of a known transcription start site and enhancers were defined as H3K4me1[+] and/or H3K27ac[+], and H3K4me3[−] regions within increasing bin sizes of 50-250 Kb upstream and downstream of the transcriptional start site in ES cell-derived cardiac progenitors and cardiomyocytes, and E14.5 fetal hearts (*Wamstad et al., 2012*). Accession numbers for histone mark data were: H3K4me1 – ENCSR000CDL; H3K27ac – ENCSR000CDK; HK4me3 – ENCSR000CDM. Variant enrichment within each feature was defined as the proportion of variants that overlapped the feature, with normalization for the proportion of coverage of the feature within the QTL 1-LOD drop-off. To test the significance of the enrichment, background controls were generated by permuting each genomic feature of DEGs and non-DEGs under QTL, respectively, by randomly selecting matched size blocks (according to the tested property) under QTL regions. This process was repeated 1000 times using bedtools shuffle (version 2.25.0). For each genomic feature, the DEG/non-DEG regions were excluded from the pool of regions used to generate random interval sets. The overlap of the variants and genomic features

of DEGs and non-DEGs under QTL was compared to the mean overlap for 1000 random interval sets and empirical p-values were calculated by dividing the number of random interval sets showing equal or greater overlap than the observed by 1000.

## Code availability

The codes used to perform linkage analysis are available at https://github.com/MahdiMoradiMarjaneh/AIL (copy archived at *Marjaneh, 2023*).

## Acknowledgements

MMM held a University of New South Wales (UNSW; Sydney, Australia) International Postgraduate Award (ID3263695) and was supported in part by the NIHR Biomedical Research Centre of Imperial College Healthcare NHS Trust (London, UK). EPK held a National Heart Foundation Clinical Fellowship (Australia). The work was supported by the National Institute of Heart Lung and Blood (USA; 1RO1HL68885-01), National Heart Foundation of Australia (G06S2575; G0050738), National Health and Medical Research Council (NHMRC, Australia) (354400, 0573732; 1074386), the New South Wales Government Ministry of Health 20:20 campaign and the Victor Chang Cardiac Research Institute Innovation Centre (funded by the New South Wales Government Ministry of Health). RPH was supported by NHMRC Australia Fellowship (0573705), Senior Principal Research Fellowship (1118576), and Investigator Grant (2008743). PCP was supported by a UNSW Sydney International Postgraduate Award (UIPA) Ph.D. Scholarship.

## Additional information

### Funding

| Funder | Grant reference number | Author |
|---|---|---|
| National Health and Medical Research Council | 0573705 | Richard P Harvey |
| National Health and Medical Research Council | 1118576 | Richard P Harvey |
| National Health and Medical Research Council | 2008743 | Richard P Harvey |
| National Heart, Lung, and Blood Institute | 1RO1HL68885-01 | Richard P Harvey |
| National Heart Foundation of Australia | G06S2575 | Richard P Harvey |
| National Heart Foundation of Australia | G0050738 | Richard P Harvey |
| National Health and Medical Research Council | 354400 | Richard P Harvey |
| National Health and Medical Research Council | 0573732 | Richard P Harvey |
| National Health and Medical Research Council | 1074386 | Richard P Harvey |
| New South Wales Government | | Richard P Harvey |
| University of New South Wales | | Paola Cornejo-Paramo |
| University of New South Wales | ID3263695 | Mahdi Moradi Marjaneh |
| Imperial College Healthcare NHS Trust | | Mahdi Moradi Marjaneh |

| Funder | Grant reference number | Author |
| --- | --- | --- |

The funders had no role in study design, data collection and interpretation, or the decision to submit the work for publication.

## Author contributions

Mahdi Moradi Marjaneh, Conceptualization, Data curation, Software, Formal analysis, Investigation, Visualization, Methodology, Writing – original draft, Writing – review and editing; Edwin P Kirk, Conceptualization, Data curation, Formal analysis, Supervision, Investigation, Visualization, Methodology, Writing – original draft, Project administration, Writing – review and editing; Ralph Patrick, Conceptualization, Data curation, Formal analysis, Investigation, Visualization, Methodology, Writing – original draft, Writing – review and editing; Dimuthu Alankarage, Conceptualization, Investigation, Visualization, Methodology, Writing – review and editing; David T Humphreys, Conceptualization, Data curation, Formal analysis, Investigation, Visualization, Methodology, Writing – review and editing; Gonzalo Del Monte-Nieto, Investigation, Methodology, Writing – review and editing; Paola Cornejo-Paramo, Formal analysis, Investigation, Methodology, Writing – review and editing; Vaibhao Janbandhu, Data curation, Investigation, Methodology, Writing – review and editing; Tram B Doan, Data curation, Supervision, Investigation, Methodology, Writing – review and editing; Sally L Dunwoodie, Supervision, Investigation, Methodology, Writing – review and editing; Emily S Wong, Chris Moran, Ian CA Martin, Conceptualization, Supervision, Investigation, Methodology, Project administration, Writing – review and editing; Peter C Thomson, Conceptualization, Supervision, Funding acquisition, Investigation, Methodology, Writing – original draft, Project administration, Writing – review and editing; Richard P Harvey, Conceptualization, Formal analysis, Supervision, Funding acquisition, Investigation, Methodology, Writing – original draft, Project administration, Writing – review and editing

## Author ORCIDs

Mahdi Moradi Marjaneh http://orcid.org/0000-0002-9412-9029
Ralph Patrick http://orcid.org/0000-0003-0956-1026
David T Humphreys http://orcid.org/0000-0003-4140-0089
Vaibhao Janbandhu http://orcid.org/0000-0002-3837-1865
Sally L Dunwoodie http://orcid.org/0000-0002-2069-7349
Emily S Wong http://orcid.org/0000-0003-0315-2942
Chris Moran http://orcid.org/0000-0003-4550-5101
Peter C Thomson http://orcid.org/0000-0003-4428-444X
Richard P Harvey http://orcid.org/0000-0002-9950-9792

## Ethics

Animals were bred and housed under Animal Care and Research Ethics approvals N00/4-2003/1/3745, N00/4-2003/2/3745, and N00/4-2003/3/3745 from the University of Sydney.

## Decision letter and Author response

Decision letter https://doi.org/10.7554/eLife.83606.sa1
Author response https://doi.org/10.7554/eLife.83606.sa2

# Additional files

## Supplementary files

• Supplementary file 1. Relationship between PFO and the quantitative traits in F14 mice with complete data (n=933).

• Supplementary file 2. Inter-trait correlation coefficients (r) in F14 mice with p-values in brackets.

• Supplementary file 3. QTL identified by AIL study.

• Supplementary file 4. High impact, deleterious missense, and deletion variants between 129T2/SvEms and QSi5 strains.

• Supplementary file 5. Splicing analysis of *Tbx20* and results of transcript isoform differential expression between strains.

• Supplementary file 6. Differentially expressed genes (DEGs) between 129T2 vs QSi5 mice across the developmental time points.

- Supplementary file 7. Gene ontology biological process terms that were significantly over-represented among the DEGs.
- Supplementary file 8. Counts of variants in different genomic features for DEGs and non-DEGs.
- Supplementary file 9. Effect of various covariates on FVL.
- Supplementary file 10. Effect of various covariates on FOW.
- Supplementary file 11. Effect of various covariates on CRW.
- Supplementary file 12. Effect of various covariates on HW.
- Supplementary file 13. Effect of various covariates on PFO.
- Supplementary file 14. List of markers with physical and genetic location.
- MDAR checklist

## Data availability

Sequencing data have been deposited in the ArrayExpress database at EMBL-EBI (https://www.ebi.ac.uk/arrayexpress) under accession codes E-MTAB-11161 (DNA-seq) and E-MTAB-10929 (RNA-seq).

The following datasets were generated:

| Author(s) | Year | Dataset title | Dataset URL | Database and Identifier |
|---|---|---|---|---|
| Moradi Marjaneh M, Kirk EP, Patrick R, Alankarage D, Humphreys DT, Del Monte-Nieto G, Cornejo-Paramo P, Janbandhu V, Doan TB, Dunwoodie SL, Wong ES, Moran C, Martin ICA, Thomson PC, Harvey RP | 2023 | DNA-seq of the QSi5 and 129T2/SvEms mouse strains | https://www.ebi.ac.uk/arrayexpress/E-MTAB-11161 | ArrayExpress, E-MTAB-11161 |
| Moradi Marjaneh M, Kirk EP, Patrick R, Alankarage D, Humphreys DT, Del Monte-Nieto G, Cornejo-Paramo P, Janbandhu V, Doan TB, Dunwoodie SL, Wong ES, Moran C, Martin ICA, Thomson PC, Harvey RP | 2023 | RNA-seq of dissected cardiac septa from a mouse developmental time course | https://www.ebi.ac.uk/arrayexpress/E-MTAB-10929 | ArrayExpress, E-MTAB-10929 |

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
