## [Editor Report]

Overall, this is a comprehensive study that will provide a useful reference for the field. It will be a useful tool for hypothesis generation, which could lead to research on therapies that target atrial septal or common congenital heart disease.

---

## [Decision Letter]

**Decision letter after peer review:**

Thank you for submitting your article "Quantitative trait and transcriptome analysis of genetic complexity underpinning cardiac interatrial septation in mice using an advanced intercross line" for consideration by *eLife*. Your article has been reviewed by 3 peer reviewers, and the evaluation has been overseen by a Reviewing Editor and David James as the Senior Editor. The reviewers have opted to remain anonymous.

Essential revisions:

All three reviewers judged this to be a comprehensive, well-designed study that will enhance our understanding of the genetics of septal defects. The reviewers' comments related primarily to the presentation of the results, in several cases requesting a further discussion of the limitations of the findings.

Reviewer 1 suggests that functional validation would greatly strengthen the manuscript. While I agree, the present results are very substantial and do provide a better overall understanding of the complexity of the trait in addition to candidates and hypotheses for future studies.

Reviewer 2 had several criticisms. I felt that points 2, 3, and 10 should be addressed, and I would be interested to know how the authors feel about all of the points. Some of the suggestions, such as the identification of rare variants that have arisen since during the development of the advanced intercross lines (point 5) would be very difficult, but the authors might expand on the point in the discussion.

Reviewer 3 also had several comments. Points 1 and 2 should be addressed. Points 3 – 5 deal with additional transcriptomic and functional analyses. I think the suggestions are excellent and the analyses would strengthen the paper, but given the extensive analyses reported I do not want to insist on additional experiments. The authors should, however, discuss these points in terms of limitations.

*Reviewer #1 (Recommendations for the authors):*

A limitation of the study in its current form is that it does not develop an experimental example of using the data set on gene regulation network analysis in Figure 7-8 to make predictions and then test them. For example, the author discusses changes to ubiquitin C, a prominent down-regulated hub gene. However, the functional significance of these predicted interactions is not experimentally tested. To demonstrate that one or more of these genes (Znrf3, Ube4a, Akt1, and Prkaca) are directly involved in the developing atrial septum would significantly increase enthusiasm for the paper.

*Reviewer #2 (Recommendations for the authors):*

1. The AIL analysis was limited to QTL previously found in F2 analysis, where the selected markers covered only a limited part of the genome. WGS and RNA seq were utilized to validate QTL genes of known role in CHDs, missing an important opportunity for identifying new QTL and novel gene discovery in PFO/CHDs.

2. Authors stated that "the possibility that FVL- and FOW- QTL could be explained by variation in HW or BW has not been formally excluded. Markers of normalized heart weight were included in AIL, like MMU11". However, in [Table 1] HW and BW were listed as independent traits. The manuscript would benefit from using HW/BW ratio and examining HW/BW association with PFO parameters, instead of using HW and BW separately as independent traits.

Furthermore, SEX and Age were not included. Adding these co-variants in their comparison between F2 and F14 in Table 1 would be important.

3. Page 13. Line 231: "The AIL results for each chromosome of quantitative traits and binary PFO are compared in figure 3". The cited figure should be [figure 4] instead of [figure 3], or [both Figure 3 and figure 4].

4. page 16, lines 277-280. "MMU9 QTL is a suggestive QTL [LOD=3.4] that was identified in F2 study and resolved into 4 QTL in the current F14 study [Figure 3 E and Figure 4 E], with significant overlapping FOW, FVL, and PFO peaks. Importantly, TBX20, a cardiac transcription factor gene involved in septal defects, is located within the 1-LOD region of the first QTL". However, only synonymous variants were found in TBX20 (based on WGS analysis) (page 20, line 368).

Given the importance of TBX20 in PFO and the suggestive data, it is worthy to list TBX20 variants and examine/discuss potential miss-classification of these variants, or potential cryptic splice site variants in TBX20 that could be missed on initial pipeline analysis.

5. Page 19. Line 340-341: "regions with low SNP density may contain rare variants that have arisen since establishment of the parent strains".

I. Why were these potentially rare variants dismissed in the current manuscript?

II. Since the prevalence of PFO in F14 increased by ~ 2 folds, compared to F2, it would be important to include the rare variants (Homozygous or heterozygous) that might have arisen since the establishment of parental strains, and ca potentially have large effects to explain the increased prevalence at F14.

6. How did the filtered deleterious variant segregate with extreme PFO phenotypes? Can they explain the higher prevalence of PFO in 129T2/SvEms (high PFO) compared to the QSi5 (low PFO) mouse strain?

7. Majority of the highlighted genes, including LRP2, SMAD6, and SIK3, are affected with 1 or 2 deleterious variants, suggesting a potential monogenic effect caused by rare variants.

I would recommend listing the minor allele frequency (MAF) of these variants and other characteristics of the variants (variant type, Sift Score, Splice prediction) in a new summary Table.

8. SMAD6 variant. The gene was identified in the QSI5-QTL (this strain exhibits low PFO). Based on the functional evidence (cell culture), this variant is damaging and may alter SMAD6 function in several ways.

I. Based on the data and discussion, it is difficult to tell whether the SMAD6 variant is contributing or protective in the PFO phenotype.

II. Again, this seems to be a monogenetic effect.

III. How does SMAD6, or its downstream effectors, contribute to the interactome network of E14.5? is it expressed in the Atrial septum during development? Does the variant influence network interactions?

9. Page 27. figure 7. PCAs for coding vs noncoding. The noncoding transcript showed a different pattern of variation (strain vs developmental stage). Since they were not included in the downstream analysis, I would suggest removing Figure 7B from the current manuscript.

10. Transcriptome analysis/Network analysis [fig7.E, Figure 8]. Adding genes affected with variants to network analysis of the DEGs may confound network structure, as no evidence was shown to support the impact of variants on their gene expression. Instead, I would suggest the following:

I. Discuss why E14 revealed the highest # of DEGs and more significant networks.

II. What was the expression status of genes with pathogenic variants that were expressed at each time point?

III. The finding that DEGs in QTL regions were enriched in pathogenic variants led to the speculation of the functional impact of some QTL on septal traits will involve collective effects of multiple variants (polygenic effect). However, the role of the list of genes carrying pathogenic variants in septum morphology has not been examined mechanistically, including smad6, in vivo. Therefore the evidence does not support the conclusion.

IV. Page 36. Line 645-654. Variants overlapping with the noncoding element.

What is the significance of this finding? Any further evidence of deleterious impact or causality?

*Reviewer #3 (Recommendations for the authors):*

1. In Figure 6, the authors filter a large number of identified variants between the AIL parental strains using a logical pipeline. The authors ultimately intersected potentially pathogenic variants under septal QTL regions with genes according to prior association with cardiovascular phenotypes (Mouse Genome Informatics; MGI) or link to human CHD from a curated list of high-confidence and emerging CHD genes. Were CHD or MGI genes enriched under QTL regions?

2. The examples of high-impact variants are highly subjective other than smad6, which the authors validate. The caveats underlying the description of unvalidated specific variants should be more explicitly noted, as few if any of them have been directly implicated in atrial septation.

3. Figure 8 – RNA-seq is performed beginning at E12.5, after the formation of the PAS, MC, and DMP. Thus, it is probably most appropriate to limit the implication of genes differentially expressed between the two mouse strains at these later time points to the formation of the septum secundum. While the formation of the SS is critical for FO closure, the limited applicability of the RNA-seq data for descriptions of atrial septation should be mentioned. Alternatively, the authors could perform additional RNA-seq experiments beginning around E9.5 in the pSHF and common atrium to include genes associated with all atrial septum structures as they form, or obtain and analyze previously published RNA-seq data from these regions.

4. The authors' analysis of the atrial septum transcriptome data jumps quickly to considering stain-specific DEGs. However, this is the first transcriptional profiling of the later stages of the atrial septum, as far as I know. A more considered baseline analysis should be performed before jumping into the strain-specific analysis. What does the transcriptome alone say about atrial septation? How does the atrial septum transcriptome compare to the atrial free wall or other regions of the heart (by comparison with RNA-seq from the authors or others)? How do the pathways that are septum-enriched compare to those uncovered in the stain-specific septum analysis?

5. Figures 7 & 8 – the authors implicate the BMP, ubiquitination, Akt, and PKA pathways in PFO based on gene expression differences between 129T2/SvEms and QSi5 mouse lines from E12.5-E16.5. These findings would be greatly strengthened if the authors could demonstrate functional differences in the activities of these pathways in atrial septum tissues in vivo at these time points. This would not be necessary for the publication of this already detailed manuscript and could be discussed as an avenue for follow-up.

6. The observation that differentially expressed QTL genes are enriched for genomic variants is an important observation. The authors used a very conservative relationship between candidate regulatory regions and genes, described as "within 50kb upstream of the transcriptional start site of relevant genes". Given that enhancers demonstrate no directional bias, their capture could be increased to include regions both upstream and downstream of a gene's TSS. Also, 50kb is quite conservative, and their capture may be increased by enlarging the considered regions to 100 or 250 kb. Given the conclusion that there is a "strong mechanistic link between variant architecture and differential expression of genes under QTL", the authors should present the gene sets and pathways that are enriched in this analysis.

7. Figure 10 – The authors could consider an analysis of variant conservation between human and mouse to this figure to interrogate the relevance of identified loci for human ASD/PFO. For variants in and around DEGs underlying QTL, what proportion in coding regions are conserved? What proportion in non-coding regions?

---

## [Author Response]

Essential revisions:Reviewer #1 (Recommendations for the authors):A limitation of the study in its current form is that it does not develop an experimental example of using the data set on gene regulation network analysis in Figure 7-8 to make predictions and then test them. For example, the author discusses changes to ubiquitin C, a prominent down-regulated hub gene. However, the functional significance of these predicted interactions is not experimentally tested. To demonstrate that one or more of these genes (Znrf3, Ube4a, Akt1, and Prkaca) are directly involved in the developing atrial septum would significantly increase enthusiasm for the paper.

Thank you for this comment. Beyond our in vitro analysis of the *Smad6* variant, we agree that additional testing of DEGs and genes carrying predicted pathogenic variants would be desirable to determine their unique contribution to atrial septal development in mouse and whether they contribute to septal defects in humans. However, I am sure the reviewers appreciate, this would require analysis of possibly many individual variants or groups of variants genetically (for example using CRISPR/Cas9). Furthermore, whereas it seems evident that biosynthetic processes affecting mitochondria, ribosomes, ECM, chromatin and ubiquitin-related processes underpin differences in atrial septal development between founder strains (a primary finding of this paper), it is not immediately obvious which individual genes are the hub driver genes that would be the most appropriate candidates for genetic manipulation. Furthermore, whereas downregulation of ubiquitin or kinases AKT and PKA, and a host of their target genes, may contribute to the defects, manipulating these genes will likely have broad and general effects on atrial cell viability, proliferation and/or interactions, and the complex interactions between variant pathways would be lost. Unravelling higher level network control would certainly be a worthy problem in cardiac system genetics, potentially relevant to other diseases involving disrupted morphogenesis; however, as noted by the handing editor, such studies are well beyond the scope of this current paper. We have added a note to the last paragraph of the Discussion (line 756) to acknowledge these issues:

“Our results provide the first high-resolution picture of genetic complexity and multilayered network liabilities underpinning atrial septal variation in a mouse model. Elucidating the impacts of genetic variants under QTL at the individual gene level would be a daunting task; however, understanding higher level network control of septal development underpinning observed defects in septal macromolecular synthesis pathways is an important cardiac developmental systems genetics problem, which may have relevance to other congenital dysmorphologies.”

Reviewer #2 (Recommendations for the authors):1. The AIL analysis was limited to QTL previously found in F2 analysis, where the selected markers covered only a limited part of the genome. WGS and RNA seq were utilized to validate QTL genes of known role in CHDs, missing an important opportunity for identifying new QTL and novel gene discovery in PFO/CHDs.

We agree with this comment and had previously address it in the Discussion (line 662). The approach taken was partly dictated by efficiencies of time and funding. The lost opportunity notwithstanding, we would argue that the approach led to important discoveries underpinning atrial septal defects, such as the strong concordance of PFO and atrial septal traits (not evident in the F2 study), the compelling protein network associated with DEGs suggesting defects in biosynthesis of macromolecular structures, and findings related to variant distribution and gene architecture.

2. Authors stated that "the possibility that FVL- and FOW- QTL could be explained by variation in HW or BW has not been formally excluded. Markers of normalized heart weight were included in AIL, like MMU11". However, in [Table 1] HW and BW were listed as independent traits. The manuscript would benefit from using HW/BW ratio and examining HW/BW association with PFO parameters, instead of using HW and BW separately as independent traits.Furthermore, SEX and Age were not included. Adding these co-variants in their comparison between F2 and F14 in Table 1 would be important.

Thank you for these interesting comments. Please note that in our previous paper on F2 mice (Kirk et al., 2006), QTL for atrial septal traits persisted after normalisation for HW, BW, sex, age, day of dissection, and coat colour. Whereas these parameters contributed to septal traits in both F2 and F14 mice, the effects were small and HW had no impact on PFO. Specifically, in the current study, the correlation coefficients (r) for comparisons between HW and BW, and septal traits, were very low (0.005 – 0.13) (Supplementary File 2). Note also that our analysis of HW QTL in F2 and F14 mice was determined after normalisation for age and sex. Importantly, our mapping results showed that HW and BW QTL were largely different from QTL for septal traits. Thus, with the caveat that mapping was not genome-wide in the F14 study, we can be very confident that QTL septal traits are not a function of variation in HW or BW. This is a very important result and, as we mention in the Discussion, HW QTL likely relates to ventricular growth as an independent parameter to development of the “primary” myocardial component of the heart from which the atrial septum derives.

We did not analyse the relationship between HW/BW ratio and PFO, nor use HW/BW ratio in QTL mapping. With respect, we do not think this effort is justified for several reasons. First, as noted above, there is only a weak contribution of HW and BW to septal traits. Furthermore, HW and BW are generally well correlated in mammals, except in pathology where there is adaptive hypertrophy and/or inflammation. The correlation coefficient between HW and BW in F14 mice was 0.64 (p<0.001; Supplementary File 2). Thus, for HW/BW ratios, these parameters will tend to cancel each other out.

To explain our initial interest in HW and BW further, QSi5 mice, which show a low prevalence of PFO (4.6%), were originally bred because they were highly fecund and showed rapid grow. Thus, they had considerably higher BW and HW than 129T2 mice. It seemed possible that genes (and even environmental factors) influencing overall growth rate would affect the size of the heart and that this would create “noise” in the specific septal traits such as FVL and FOW, which correlate with PFO. Thus, HW and BW were included in analyses of the septal traits not because we were interested in them per se, but because we wished to quantify their influences, hoping to formally exclude them as a trivial source of septal variation. Specifically, we wanted normalised measures of FVL, FOW etc independent of variation in HW and BW. After retrospective analysis of QTL for HW in F2 data revealed a number of HW (and BW) QTL, it was simple enough also to include HW as a variable in selection of F14 mice for QTL analysis of septal traits. This analysis of HW QTL allowed us to show that they largely do not overlap with FVL and FOW QTL, providing much stronger evidence that HW is not a significant confounder in our analysis of the genetic basis of atrial septal defects in 129T2 mice.

Finally, our quantitative genetics team (CM, ICAM, PCT) point out that most statisticians would shy away from using ratios rather than adjusted values in QTL analyses. That is, there is no real certainty that having a term β×HW/BW in the model would capture variation any better than β_1_×HW + β_2_×BW, particularly because the point is to adjust for the source of variation in estimating QTL effects, as noted above, not to explore the underlying functional form of any relationship. There is a long history of objections by statisticians to the use of ratios going back to Pearson (1897), with Kronmal 1993 (https://doi.org/10.2307/2983064) and Curran-Everett 2013 (https://doi.org/10.1152/advan.00053.2013) providing theoretical justifications.

In summary, whereas differential growth between mouse strains is clearly an interesting topic, and may have translational relevance in animal breeding, this is not the topic of this paper and would represent an unhelpful distraction from the primary focus of this work on septal traits. We hope this is acceptable to the reviewers. We have, however, clarified further specifically why HW and BW were of interest in the Results section “Selection of AIL mice – heart weight and body weight phenotypes” (line 198):

“Given that QSi5 mice were originally selected for their high fecundity and growth, it was evident that the quantitative septal parameters under study might be influenced by heart size and mass (Table 1) (Kirk et al., 2006), and indeed both FVL and FOW were significantly correlated with HW in both F2 and F14 cohorts, albeit that the effects were small (Supplementary File 2) (Kirk et al., 2006) and HW had no influence the likelihood of PFO.”

3. Page 13. Line 231: “The AIL results for each chromosome of quantitative traits and binary PFO are compared in figure 3”. The cited figure should be [figure 4] instead of [figure 3], or [both Figure 3 and figure 4].

Thank you, Figure 3 has been corrected to Figure.

4. page 16, lines 277-280. "MMU9 QTL is a suggestive QTL [LOD=3.4] that was identified in F2 study and resolved into 4 QTL in the current F14 study [Figure 3 E and Figure 4 E], with significant overlapping FOW, FVL, and PFO peaks. Importantly, TBX20, a cardiac transcription factor gene involved in septal defects, is located within the 1-LOD region of the first QTL". However, only synonymous variants were found in TBX20 (based on WGS analysis) (page 20, line 368).Given the importance of TBX20 in PFO and the suggestive data, it is worthy to list TBX20 variants and examine/discuss potential miss-classification of these variants, or potential cryptic splice site variants in TBX20 that could be missed on initial pipeline analysis.

Thank you for this suggestion and we agree. We have modified the text first to be more specific about the number of variants detected across the *Tbx20* gene, which we now list in a new Supplementary File 5. There were 321 variants in total – only 3 were exonic variants and these were synonymous as previously indicated. The rest were intronic; however, the *VEP* tool that we used for variant classification detected no variants overlapping known splice motifs and no cryptic splice motifs. We reran the analysis using *Spliceogen*, a recently described tool pipeline that integrates some of the best performing models for splice motif prediction and achieved the same results. Again, no splicing defects were detected. *Spliceogen* provides a ranking for cryptic donor/acceptor sites, however, these do not reflect real probabilities. We therefore analysed splice junctions detected in the RNA-seq but did not find confident differences between the strains. In addition, we used *Salmon* to map *Tbx20* isoforms and perform differential expression analysis, but this approach also did not reveal differences in *Tbx20* isoforms between strains. For future reference, all the above outputs are given in Supplementary File 5. We have updated the methods and inserted the following in the manuscript (line 383):

“For the Tbx20 gene on MMU9, a total of 321 variants were detected (Supplementary File 5); however, only three were exonic and these were synonymous. No defects in splice motifs or cryptic splice donor/acceptor sites were detected using the VEP tool. We reran the analysis using Spliceogen (*Monger et al., 2019*) and again no splicing defects were detected. Spliceogen provides rankings of possible cryptic splice sites (Supplementary File 5), however, these do not reflect actual probabilities. Therefore, we analysed splice junctions detected in RNA-seq data from the developing septum (see below) but did not find splice site differences for Tbx20 between parental strains. We also mapped Tbx20 transcript isoforms with Salmon (*Patro et al., 2017*) and performed differential expression analysis, again yielding no differences in isoform expression between strains (Supplementary File 5).”

5. Page 19. Line 340-341: "regions with low SNP density may contain rare variants that have arisen since establishment of the parent strains".I. Why were these potentially rare variants dismissed in the current manuscript?II. Since the prevalence of PFO in F14 increased by ~ 2 folds, compared to F2, it would be important to include the rare variants (Homozygous or heterozygous) that might have arisen since the establishment of parental strains, and ca potentially have large effects to explain the increased prevalence at F14.

Thank you for these comments. We think that use of the term “rare variants” when referring to variants that might have arisen in low SNP density regions since the establishment of the parental strains is confusing because it is generally used to refer to variants with low MAF in humans. However, they will be “rare” in the sense that if they did arise in the germline, they would tend to be lost by genetic drift. We neglected to acknowledge that they might also arise in regions of high SNP density as well. We have corrected this by referring to them as simply “variants” (line 352) and we have also modified the text to acknowledged that variants that have arisen since establishment of parental strains could in principle also contribute to septal defects:

“Regions with a high rate of polymorphism between parental strains (spanning nearly one-third of the genome but harboring more than 95% of the genetic variation) can significantly reduce the regions of interest underlying QTL (Wade et al., 2002), albeit that regions with high or low polymorphism may also carry variants that have arisen since the establishment of parent strains (Bloom et al., 2019). Although these would be rare, they could, in principle, also contribute to septal variation.”

Please note that in filtering for pathogenic coding variants, variants lying within regions of low SNP density were not excluded. Predicted pathogenic variants shown in Figures 6A,B are annotated with respect to their presence in high (starred) or low SNP density regions. We have now indicated in the text also the number and percent of filtered variants which lie in high density SNP regions (84/92; 91.3%; line 380) and the status of each variant is now also indicated in Supplementary File 4.

The 2-fold increase in PFO prevalence in F14 mice compared to F2 mice is interesting; however, the most obvious explanation for this is genetic drift during the cascade breeding of F14 mice. We think the increased PFO prevalence is unlikely to relate specifically to variants that have arisen since establishment of the parental strains for the reasons above and we have not attempted to focus specifically on such variants – we don’t believe that there is any a priori reason for doing this. As noted, predicted pathogenic variants in high (starred) and low density SNP regions are shown in Figure 6A,B.

6. How did the filtered deleterious variant segregate with extreme PFO phenotypes? Can they explain the higher prevalence of PFO in 129T2/SvEms (high PFO) compared to the QSi5 (low PFO) mouse strain?

Thank you for this comment. It is in fact difficult to say whether filtered variants “explain” the extreme PFO phenotypes, noting that cis-regulatory variants will likely be extremely important also, as we highlight later in the manuscript. Nonetheless, reflecting on the Reviewer’s comment, we reexplored the effect sizes for QTL under which the high-confidence pathogenic variants highlighted in the text lay (i.e. those predicted pathogenic *and* overlapping curated MGI and CHD lists). These variants were selected based principally on due diligence of the predicted impact of the variants (or multiple variants) and/or the involvement of the relevant gene in heart development and/or disease. We discovered that QTL for most highlighted variants had large effect sizes (*Cep164*: 65%; *Dst*: 5.68%; *Sik3*: 65%; *Lrp2*: 30%; *Smad6*: 58%). However, we noted that all bar one of these variants were associated with cryptic QTL (i.e. protective of septal defects), including the *Smad6* variant. We also found that variants were often found in the unexpected strain give the direction (normal or cryptic) of the QTL. This highlights that there are highly complex interactions between alleles within and between QTL in parental strains and in individual AIL mice. At this stage, we prefer not to make assumption about the network model, which contains multiple interacting elements, albeit a default model where one variant is responsible for each QTL, seems naïve. We feel that trying to segregate highlighted variants with extremes of phenotypes will likely lead us for the moment down an unproductive path. To provide more clarity in the text and follow on from the new findings, we have indicated in the Results section the effect size of the QTL under which the highlighted variants lay (pages 18-20) and added a section in the Discussion about possible implications of our findings for the genetic model (pages 33-34).

“Our results are consistent with Fisher's 'infinitesimal' model whereby many loci (theoretically an infinite number) could each contribute to a small part the liability for common disease (*Fisher, 1918*), and also GWAS analysis of human disease that most often detect common risk loci of small effect. However, as in our previous study (*Kirk et al., 2006*), individual septal QTL can have significant effect sizes, demonstrating the power of the AIL approach to discern alleles of major impact. Indeed, all five predicted pathogenic coding variants highlighted in this study, including the Smad6 variant, lay under QTL of moderate to high effect size (5.68-65%). It is possible that a limited number of high impact variants are responsible for septal defects in certain AIL mice. This type of polygenic inheritance has been demonstrated recently in a human cardiomyopathy family (*Gifford et al., 2019*). Such variants may prove to have causative or modifying effects on atrial septal defects in humans, as exemplified by SMAD1.

Several findings speak to the complex genetic landscape of PFO. It is interesting that twelve out of the 37 QTL detected were cryptic QTL, with many being of moderate to large effect size (~5-70%; Supplementary File 3). Thus, cryptic QTL are frequently detected and individually appear to provide strong buffering effects against septal defects. Cryptic QTL have been detected commonly in other animal and plant QTL studies (*Rieseberg et al., 1999*). The mechanisms of their action would be interesting to identify as they may reveal pathways that protect against CHD more broadly.

A further complexity is that, of five genes carrying predicted pathogenic variants highlighted in this study, four were found in the unexpected strain. For example, Cep164 lies under a cryptic QTL (protective), whereas predicted pathogenic variants in Cep164 were found in the 129T2/SvEms strain (showing worse septal status). The inverse can also apply – for example, Lrp2 lies under a normal QTL, but variants were found in the QSi5 strain. Such findings suggest highly complex interactions within and between individual QTL. As noted, 26% of predicted pathogenic genes, some known to be involved in heart development, carried multiple predicted pathogenic variants (Cep164 carried seven). It would be reasonable to imagine that these alleles are null or have major impacts on gene function. Genetic compensation may also buffer the impact of such variants (*El-Brolosy et al., 2019*).”

7. Majority of the highlighted genes, including LRP2, SMAD6, and SIK3, are affected with 1 or 2 deleterious variants, suggesting a potential monogenic effect caused by rare variants.I would recommend listing the minor allele frequency (MAF) of these variants and other characteristics of the variants (variant type, Sift Score, Splice prediction) in a new summary Table.

We agree that the function of genes predicted as pathogenic and carrying multiple variants may be severely compromised. See response to Comment 6 above. Please note that the general classification of variants (High Impact; Deleterious [SIFT]; non-frameshift deletion) is recorded in Supplementary File 4.

8. SMAD6 variant. The gene was identified in the QSI5-QTL (this strain exhibits low PFO). Based on the functional evidence (cell culture), this variant is damaging and may alter SMAD6 function in several ways.I. Based on the data and discussion, it is difficult to tell whether the SMAD6 variant is contributing or protective in the PFO phenotype.II. Again, this seems to be a monogenetic effect.III. How does SMAD6, or its downstream effectors, contribute to the interactome network of E14.5? is it expressed in the Atrial septum during development? Does the variant influence network interactions?

*Smad6* is expressed in the developing atrial septum and the *Smad6* variant is an excellent candidate for being protective against septal defects (i.e. underlies a cryptic QTL and appears in the QSi5 strain with robust septal qualities). We clarify this at the end of the *Smad6* section on page 23:

“Because the Smad6 variant lies under a cryptic QTL of high effect (58%) and is seen in the QSi5 strain (with robust septal qualities), it is an excellent candidate for being protective against atrial septal defects. We confirmed Smad6 expression in the developing interatrial septum (see below).”

It is unclear precisely how the variant will affect SMAD6 function (and therefore control of the BMP pathway) in vivo, noting that it still has repressive activity on the BMP pathway assay in vitro. However, it is possible that it may be an allele of strong effect (we avoid the term “monogenic” as it would be difficult to claim this, even for alleles of strong effect). We currently don’t know how it affects the differential expression of septal genes between strains and associated protein:protein interaction networks.

9. Page 27. figure 7. PCAs for coding vs noncoding. The noncoding transcript showed a different pattern of variation (strain vs developmental stage). Since they were not included in the downstream analysis, I would suggest removing Figure 7B from the current manuscript.

We believe the reviewer is referring to PCA plot using non-coding genes in Figure 8B. As suggested, this has been removed and numbering of other Figure 8 panels and text adjusted accordingly.

10. Transcriptome analysis/Network analysis [fig7.E, Figure 8]. Adding genes affected with variants to network analysis of the DEGs may confound network structure, as no evidence was shown to support the impact of variants on their gene expression. Instead, I would suggest the following:

We agree that the presence of pathogenic protein coding variant (red nodes) in the E14.5 STRING network is not helpful, so have reconfigured the networks without them and adjusted the text and legend accordingly. Deleting the red nodes did not have a major effect on the networks (new Figure 8 and Figure 8—figure supplements 1 and 2). We also inserted a note at the end of this section on page 25 highlighting the signalling kinase SRC, which forms a strong hub at E12.5, and also that there were very few DEGs that contained predicted pathogenic coding variants:

“Interestingly, along with hubs for Ubc, collagen/ECM and translation, an additional hub centred on Src, encoding a proto-oncogene tyrosine kinase involved in embryonic development and cell growth, was also evident at E12.5, but not at later stages (Figure 8—figure supplement 1). Across all stages, there were very few DEGs that were predicted to carry pathogenic coding variants, and none appeared as hub genes, placing network genes downstream of functional variants.”

I. Discuss why E14 revealed the highest # of DEGs and more significant networks.

Thank you for challenging us with this question. It is possible that the most significant STRING network of DEGs between strains occurs at E14.5 (subnetworks for macromolecular synthesis of mitochondria, chromatin, ribosomes and ECM) because changes relate to growth and maturation of septal components at a relatively later stage of septal development rather than to specific (earlier) steps of septal component specification. This notion would be supported by the fact that there is strong overlap between QTL for FVL, FOW and PFO, and other findings. We have made a comment to this effect in the Discussion on page 32 (line 694):

“By far the most significant STRING protein network for DEGs was at E14.5, and subnetworks with high connectivity relating to macromolecular cellular structures including nucleosomes, ribosomes, mitochondria and ECM, were detected. Virtually all associated genes were downregulated in the 129T2/SvEms strain (showing less robust septal characteristics). Our findings suggest that there is a general septal growth and maturation deficit involving multiple septal components in the 129T2/SvEms strain at a relatively late stage (E14.5) of septal development. This idea is supported by the correlations between FVL and FOW (reflecting development of the primum and secundum septa, respectively) (Kirk et al., 2006) and the strong concordance between QTL for individual septal traits and PFO discovered in this work. We hypothesise that differences in quantitative septal traits and PFO prevalence that we documented previously in inbred, hybrid and mutant strain of mice (Kirk et al., 2006), relate in part to degrees of septal growth and maturation, and that higher septal growth and maturation facilitates better fusion of the septum primum and secundum. Careful elucidation of the differential growth parameters of septal components in parent strains across time may provide further support for this model.”

We think this is an important hypothesis that integrates diverse findings and so we have incorporated a brief reference to it in the Abstract.

“Transcriptome analysis of developing septa revealed downregulation of networks involving ribosome, nucleosome, mitochondrial and extracellular matrix biosynthesis in the 129T2/SvEms strain, potentially reflecting an essential role for growth and cellular maturation in septal development.”

II. What was the expression status of genes with pathogenic variants that were expressed at each time point?

We have added a new tab in Supplementary File 4 indicating raw expression figures for all predicted pathogenic genes at each stage of atrial septal development analysed, with a column indicating whether the gene was differentially expressed. Around 30% of genes were differentially expressed in at least one time point and we have added a note to this effect on page.

III. The finding that DEGs in QTL regions were enriched in pathogenic variants led to the speculation of the functional impact of some QTL on septal traits will involve collective effects of multiple variants (polygenic effect). However, the role of the list of genes carrying pathogenic variants in septum morphology has not been examined mechanistically, including smad6, in vivo. Therefore the evidence does not support the conclusion.IV. Page 36. Line 645-654. Variants overlapping with the noncoding element.What is the significance of this finding? Any further evidence of deleterious impact or causality?

We were confused by statement under III that DEGs under QTL are not enriched in pathogenic variants. We think the Reviewer was referring to DEGs under QTL that were over-represented in variants (129T2 vs QSi5) which lie in non-coding features. We have not tested the “polygenic” hypothesis mechanistically. We address the substantial challenges in doing so in our response to Reviewer 1.

Reviewer #3 (Recommendations for the authors):1. In Figure 6, the authors filter a large number of identified variants between the AIL parental strains using a logical pipeline. The authors ultimately intersected potentially pathogenic variants under septal QTL regions with genes according to prior association with cardiovascular phenotypes (Mouse Genome Informatics; MGI) or link to human CHD from a curated list of high-confidence and emerging CHD genes. Were CHD or MGI genes enriched under QTL regions?

We in fact made an error in our reporting in the Results section on how many of the filtered (predicted pathogenic) variants overlapped with the curated MGI and CHD lists of pathogenic genes. We have now corrected this and, as shown in the Venn diagrams of Figure 6C,D, eight predicted pathogenic genes overlapped the MGI/CDH lists. We list these in the text on page and again in the Discussion on page. As requested, we tested whether curated genes were overrepresented under QTL (relative to non-QTL, atrial-expressed genes), and they were not. This does not diminish their potential involvement in atrial septal traits; however, this has not been tested.

2. The examples of high-impact variants are highly subjective other than smad6, which the authors validate. The caveats underlying the description of unvalidated specific variants should be more explicitly noted, as few if any of them have been directly implicated in atrial septation.

We understand the Reviewer’s comment and have amended to text at the end of the section (page 490) to indicate that variants other than *Smad6* have not been validated. We also go further into the complexity of the highlighted variant’s QTL effect size, direction (normal versus cryptic) and strain of origin in the Discussion, and implications for the model (see response to Reviewer #2 comment 6).

3. Figure 8 – RNA-seq is performed beginning at E12.5, after the formation of the PAS, MC, and DMP. Thus, it is probably most appropriate to limit the implication of genes differentially expressed between the two mouse strains at these later time points to the formation of the septum secundum. While the formation of the SS is critical for FO closure, the limited applicability of the RNA-seq data for descriptions of atrial septation should be mentioned. Alternatively, the authors could perform additional RNA-seq experiments beginning around E9.5 in the pSHF and common atrium to include genes associated with all atrial septum structures as they form, or obtain and analyze previously published RNA-seq data from these regions.

Thank you for raising this. First, we have corrected out statement in the Introduction, where we had indicated that the *septum secundum* was an infolding of the dorsal atrial myocardium (as was described in human hearts). We realise now that the situation in mouse is different and that the *septum secundum* is a true septum (muscular ridge). We have clarified this in the Introduction on line 91:

“A septum secundum forms as an infolding of the dorsal atrial wall in humans (Anderson et al., 2003) or as specific muscular ridge in mice (Briggs et al., 2012), leaving an additional, prominent and offset interatrial communication termed the foramen ovale (Burns et al., 2016), completing the flap valve apparatus (Figure 1A).”

The Reviewer is correct in stating that at the earliest timepoint analysed (E12.5), the septum primum mesenchymal cap has already fused with the atrioventricular canal cushion, and the *ostium secundum* has already formed (Briggs et al., 2012). We have clarified this in the text on line 497:

“At the earliest time point, E12.5, the thin septum primum has already fused with the AV septum via its mesenchymal cap, and the ostium secundum has been created by cell death; thus, transcriptome changes across the three stages analysed may reflect prior changes during septum primum formation, as well as formation of the septum secundum, septal growth and remodelling.”

4. The authors' analysis of the atrial septum transcriptome data jumps quickly to considering stain-specific DEGs. However, this is the first transcriptional profiling of the later stages of the atrial septum, as far as I know. A more considered baseline analysis should be performed before jumping into the strain-specific analysis. What does the transcriptome alone say about atrial septation? How does the atrial septum transcriptome compare to the atrial free wall or other regions of the heart (by comparison with RNA-seq from the authors or others)? How do the pathways that are septum-enriched compare to those uncovered in the stain-specific septum analysis?

Thanks for this comment and we agree that deeper analysis of this data set could be fruitful. However, we have not undertaken such analyses so as not to distract from the main points of this already dense paper.

5. Figures 7 & 8 – the authors implicate the BMP, ubiquitination, Akt, and PKA pathways in PFO based on gene expression differences between 129T2/SvEms and QSi5 mouse lines from E12.5-E16.5. These findings would be greatly strengthened if the authors could demonstrate functional differences in the activities of these pathways in atrial septum tissues in vivo at these time points. This would not be necessary for the publication of this already detailed manuscript and could be discussed as an avenue for follow-up.

We have made a comment to this effect in the Discussion on line 711:

“We propose that variants in the 129T2/SvEms parent strain depress growth-related signalling pathways (including BMP, AKT, PKA and SRC) and associated ubiquitin-related quality control processes, impacting septal growth and morphogenesis independently of chamber growth. In this light, it is interesting that CHD and cancer risk genes have recently been correlated (Morton et al., 2021). This could be a focus area for further investigation.”

6. The observation that differentially expressed QTL genes are enriched for genomic variants is an important observation. The authors used a very conservative relationship between candidate regulatory regions and genes, described as "within 50kb upstream of the transcriptional start site of relevant genes". Given that enhancers demonstrate no directional bias, their capture could be increased to include regions both upstream and downstream of a gene's TSS. Also, 50kb is quite conservative, and their capture may be increased by enlarging the considered regions to 100 or 250 kb. Given the conclusion that there is a "strong mechanistic link between variant architecture and differential expression of genes under QTL", the authors should present the gene sets and pathways that are enriched in this analysis.

We have now expanded the analysis to include enhancer bin sizes of 50Kb (as used previously), 100Kb, 200Kb and 250Kb. For this new analysis, the definition of a bin was modified to include enhancers detected within regions (e.g. 50Kb) upstream and downstream of the transcriptional start site. The over-representation of variants in enhancers of DEGs under QTL fell within the p<0.05 threshold for all bin sizes. Please note that we detected an error in the previous version of the manuscript with respect to the labelling of one of the histone mark files we referenced (Wamstad et al., 2012) – the correct accession number is now indicated in Materials and methods. Results are presented in Figure 10—figure supplement 2 and we wrote into the text as follows (line 579):

“After permutation testing, we observed a significant enrichment of variants relative to expected in enhancers, promoters, 5UTRs and introns for DEGs underlying QTL (Figure 10). Non-DEGs showed significant depletion of variants in exons. DEGs and non-DEGs showed similar normalised variant count distributions across features (Figure 10—figure supplement 1), indicating that DEGs were not substantially skewed towards genes with high variant density. Likewise, variants were not focused on any specific gene class (Supplementary File 8). The over-representation of variants was consistent when candidate cardiac enhancers near DEGs were considered even up to 250Kb from the TSS (Figure 10—figure supplement 2A), noting however that most enhancers were found within the 100Kb window (Figure 10—figure supplement 2B). Interestingly, DEGs had approximately 60-80% more enhancers than non-DEGs, consistent with their higher levels of expression, irrespective of septal stage or strain (Figure 10—figure supplement 2C-E).”

With regards to “presenting the gene sets and pathways that are enriched in this analysis”, thank you for this suggestion. This is of course difficult because of the statistical nature of the tests. However, we ranked DEGs under QTL based on the variant density or binary variant status across enhancers, promoters, 5’UTRs and/or exons, and applied *fgsea* to the ranked gene lists to identify whether there was enrichment of GO terms. However, there were no significant GO terms identified. We did not proceed further with this approach.

7. Figure 10 – The authors could consider an analysis of variant conservation between human and mouse to this figure to interrogate the relevance of identified loci for human ASD/PFO. For variants in and around DEGs underlying QTL, what proportion in coding regions are conserved? What proportion in non-coding regions?

With respect, this analysis has a statistical basis, and until more specific causative coding and cis-regulatory variants can be mapped, this would be a distraction from the main discovery in this section.